# Reverse evolution leads to genotypic incompatibility despite functional and active site convergence

Miriam Kaltenbach[1,2], Colin J Jackson[3], Eleanor C Campbell[3], Florian Hollfelder[2], Nobuhiko Tokuriki[1]*

[1]Michael Smith Laboratories, University of British Columbia, Vancouver, Canada; [2]Department of Biochemistry, University of Cambridge, Cambridge, United Kingdom; [3]Research School of Chemistry, Australian National University, Canberra, Australia

**Abstract** Understanding the extent to which enzyme evolution is reversible can shed light on the fundamental relationship between protein sequence, structure, and function. Here, we perform an experimental test of evolutionary reversibility using directed evolution from a phosphotriesterase to an arylesterase, and back, and examine the underlying molecular basis. We find that wild-type phosphotriesterase function could be restored ($>10^4$-fold activity increase), but via an alternative set of mutations. The enzyme active site converged towards its original state, indicating evolutionary constraints imposed by catalytic requirements. We reveal that extensive epistasis prevents reversions and necessitates fixation of new mutations, leading to a functionally identical sequence. Many amino acid exchanges between the new and original enzyme are not tolerated, implying sequence incompatibility. Therefore, the evolution was phenotypically reversible but genotypically irreversible. Our study illustrates that the enzyme's adaptive landscape is highly rugged, and different functional sequences may constitute separate fitness peaks.

*For correspondence: tokuriki@msl.ubc.ca

**Competing interests:** The authors declare that no competing interests exist.

## Introduction

The controversy surrounding evolutionary reversibility pertains to one of the fundamental questions in evolutionary biology: the extent to which selection pressure determines evolutionary outcomes (*Teotonio and Rose, 2001*; *Gould, 2007*; *Collin and Miglietta, 2008*; *Lobkovsky and Koonin, 2012*). Also, through understanding reversibility on the levels of both phenotype and genotype, one could catch a glimpse at the structure of the respective fitness (or adaptive) landscape. The extent of ruggedness of adaptive landscapes—that is, the prevalence of epistasis, and thus historical contingency—have recently received considerable attention (*Whitlock et al., 1995*; *Poelwijk et al., 2007*; *de Visser et al., 2011*; *Breen et al., 2012*; *Harms and Thornton, 2013*; *McCandlish et al., 2013*; *Kaltenbach and Tokuriki, 2014*). While the unlikelihood of reversing a historical pathway taken by evolution has been demonstrated (*Bridgham et al., 2009*), a large number of sequences can encode functionally identical proteins ('genotypic redundancy') and phenotypic reversion can still occur via alternative pathways (*Clarke, 1985*; *Lenski, 1988*; *Crill et al., 2000*; *Teotonio and Rose, 2000*; *Kitano et al., 2008*). Yet, the evolutionary dynamics underlying phenotypic reversion have not been addressed. Does phenotypic reversion lead back to the ancestral peak on the adaptive landscape or to a new peak (*Carneiro and Hartl, 2010*; *Lobkovsky and Koonin, 2012*)? In other words, to what extent are the sequences of the ancestral and reverse-evolved proteins accessible via a neutral network—that is, are amino acid exchanges between the two proteins tolerated or result in loss of function? The inability to exchange amino acids between homologous proteins due to epistasis represents 'genotypic incompatibility' and can result in a non-functional enzyme, a phenomenon

**eLife digest** Enzymes in bacteria and other organisms are built following instructions contained within each cell's DNA. Changes in the DNA, that is to say, mutations, can alter the shape and activity of the enzymes that are produced, which can ultimately affect the ability of the organism to survive and reproduce. Mutations that are beneficial to the organism are more likely to be passed on to future generations, which can lead to populations changing over time.

The DNA sequences that an organism carries are referred to as its 'genotype' and the resulting physical characteristics of the organism are known as its 'phenotype'. Studies of evolution tend to focus on how particular species or molecules become more different over time. However, one area that remains controversial is whether it is possible for evolution to be reversed so that an organism or molecule returns to a previous form.

An enzyme called PTE is said to have phosphotriesterase activity because it catalyzes this particular type of chemical reaction. Recently, a group of researchers used a method called 'directed evolution' to demonstrate that it is possible for PTE to evolve in a way that means it loses its phosphotriesterase activity and becomes able to catalyze a different type of chemical reaction. Here, Kaltenbach et al.—including some of the researchers from the previous work—investigated whether it was possible to use the same method to reverse this evolution and restore the enzyme's original activity.

The experiments show that reverse evolution is possible as phosphotriesterase activity was restored to the PTE enzyme from the previous study. However, although the phenotype of the final enzyme matched that of the original PTE enzyme, the genotypes did not match as the DNA sequences of the genes that encode these enzymes differ. The DNA does not revert to its original sequence because the effect of individual mutations on the phenotype depends on what other mutations are present. For example, as the enzyme evolved its new activity, additional mutations accumulated that did not alter enzyme activity. During the reverse evolution experiment, some of these mutations could have started to exert influence on the phenotype so that different mutations were required to restore the phosphotriesterase activity.

In the future, Kaltenbach et al.'s findings may aid efforts to engineer artificial enzymes for use in medicine or industry.

which can be compared to the 'Dobzhansky-Muller effect' of hybrid incompatibility (*Orr, 1995*; *Kondrashov et al., 2002*).

Another important aspect to be explored is the underlying molecular mechanism of phenotypic reversibility. Restoration of function can either be brought about by the same structure and mechanism as in the ancestor, or by a distinct, alternative state. Structural convergence would indicate that functional requirements exist, which deterministically lead to one particular structural solution. On the other hand, structural divergence would imply the accessibility of various solutions that can bring about efficient catalysis. Thus, understanding the molecular basis for (ir)reversibility and (in)compatibility would provide valuable insights into protein sequence-function-structure relationships. What are the molecular requirements for a specific function? What structural changes are required to switch from one function to another? Identifying such changes, which are often based on subtle effects (e.g., on mutations occurring in remote locations, or mutations which only show a favorable effect in combination), remains a great challenge in protein science. What is the molecular basis underlying mutational epistasis, which leads to alternative evolutionary outcomes?

Directed evolution is a powerful tool to address these questions and explore adaptive landscapes because it allows the study of evolution in a highly controlled setup (*Peisajovich and Tawfik, 2007*; *Romero and Arnold, 2009*; *Kawecki et al., 2012*). High selection pressure can prevent fixation of neutral, functionally irrelevant mutations, resulting in an adaptive trajectory without mutational noise. All evolutionary intermediates (the 'molecular fossil record') are obtained, so the evolutionary dynamics and their molecular basis can be characterized in detail. Performing evolution in both the forward and reverse direction and comparing the changes in each direction provides a unique handle for identifying such effects. Understanding these phenomena would improve our ability to design and engineer novel proteins in the laboratory.

Here, we experimentally test the reversibility of enzyme evolution and investigate its molecular basis. We previously evolved the enzyme PTE, a phosphotriesterase, into an arylesterase (*Roodveldt and Tawfik, 2005*; *Tokuriki et al., 2012*; *Wyganowski et al., 2013*). In this work, we applied a selection pressure to restore the original phosphotriesterase activity. We characterized the entire trajectory including both the forward and reverse process in terms of phenotypic reversibility (function or enzymatic activity), genotypic irreversibility (sequence), as well as in terms of the underlying molecular basis (structure and mechanism). We find that PTE has a rugged adaptive landscape on which the accessibility of functional mutations is severely limited, and describe the mechanisms that lead to genotypic irreversibility and incompatibility.

## Results

### Phenotypic reversibility in the laboratory evolution of PTE

We previously reported the laboratory evolution of PTE (*wt*PTE) into a highly efficient arylesterase for 2-naphthyl hexanoate (2NH) (*Roodveldt and Tawfik, 2005*; *Tokuriki et al., 2012*; *Wyganowski et al., 2013*). In the course of the trajectory, the original phosphotriesterase activity decreased drastically ($10^4$-fold) although no selection pressure was applied against it. In this work, we first completed the functional transition by further decreasing the remaining phosphotriesterase activity ($\sim$10-fold) by four additional rounds of directed evolution for maintaining arylesterase but reducing phosphotriesterase activity (*Supplementary file 1*). Briefly, libraries were generated by error-prone PCR and transformed into *Escherichia coli* (BL21 (DE3)). As a pre-screen for arylesterase activity, protein expression was induced in the bacterial colonies on agar plates, and a mixture of the substrate 2NH and a product stain (Fast Red) was added as previously described (*Figure 1A*) (*Roodveldt and Tawfik, 2005*; *Tokuriki et al., 2012*; *Wyganowski et al., 2013*). Upon hydrolysis of 2NH, Fast Red forms a red complex with the naphtholate leaving group, meaning colonies that develop a red color contain active arylesterase variants. In each round, 2000–10,000 colonies were screened in this fashion, theoretically covering most single point mutations in the 330 amino acid PTE gene. Positive colonies were then re-grown and re-assayed in 96-well plates and initial rates of both 2NH and paraoxon hydrolysis were determined in clarified lysate. In our experience, activity increases >1.3-fold compared to the respective parent yielded reliably improved variants. The variant with the largest improvement in initial rate was then used as the template for the next round of error-prone PCR or several variants were subjected to DNA shuffling. To buffer the destabilizing effects of functional mutations and minimize reductions in soluble protein expression levels, we used GroEL/ES overexpression as previously described (*Supplementary file 1*) (*Tokuriki and Tawfik, 2009*; *Wyganowski et al., 2013*). In total, with 22 rounds of 'forward evolution', the accumulation of 26 mutations from *wt*PTE resulted in a highly efficient and specialized arylesterase (AE) with a $\sim$10$^5$-fold increase in arylesterase rates ($k_{cat}/K_M$ for 2NH >10$^6$ M$^{-1}$s$^{-1}$) and an overall $\sim$10$^5$-fold decrease in phosphotriesterase activity ($k_{cat}/K_M$ for paraoxon $\approx$10$^2$ M$^{-1}$s$^{-1}$, *Figure 1B,C*). Because selection was specific for arylester hydrolysis until round 18, the change in phosphotriesterase activity was stochastic: many mutations decreased phosphotriesterase activity (11 mutations), some were neutral (nine mutations), and others increased phosphotriesterase activity (six mutations). Starting from AE, we then performed the reverse evolution to restore phosphotriesterase activity using an experimental setup equivalent to the forward process (*Figure 1A*) with the following modifications: the pre-screen was carried out using a fluorogenic phosphotriester as a surrogate for paraoxon (*Supplementary file 1*) and then validated in 96-well format as described above. The selection criterion was now an increased initial rate of paraoxon hydrolysis. In our evolutionary model system, variant fitness is defined as the level of enzymatic activity in cell lysate. All variants were also purified and the kinetic parameters determined, which correlated well with lysate activity (*Figure 1—figure supplement 1*, *Supplementary file 2*).

The restoration of phosphotriesterase activity in the reverse evolution followed a pattern similar to that observed for arylesterase activity in the forward evolution: increasing smoothly and gradually through the stepwise accumulation of mutations (*Figure 1B,C*). Moreover, it followed a 'diminishing returns' pattern characteristic for the development of a function under selection—that is, the activity gain per mutation gradually decreased in later stages of the functional transition, where fitness reached a plateau (*Figure 1B*) (*Stebbins, 1944*; *MacLean et al., 2010*; *Chou et al., 2011*; *Khan et al., 2011*; *Tokuriki et al., 2012*). Furthermore, trade-offs between the two activities were weak in the early rounds of evolution, resulting in a generalist, bifunctional intermediate (*Aharoni et al., 2005*;

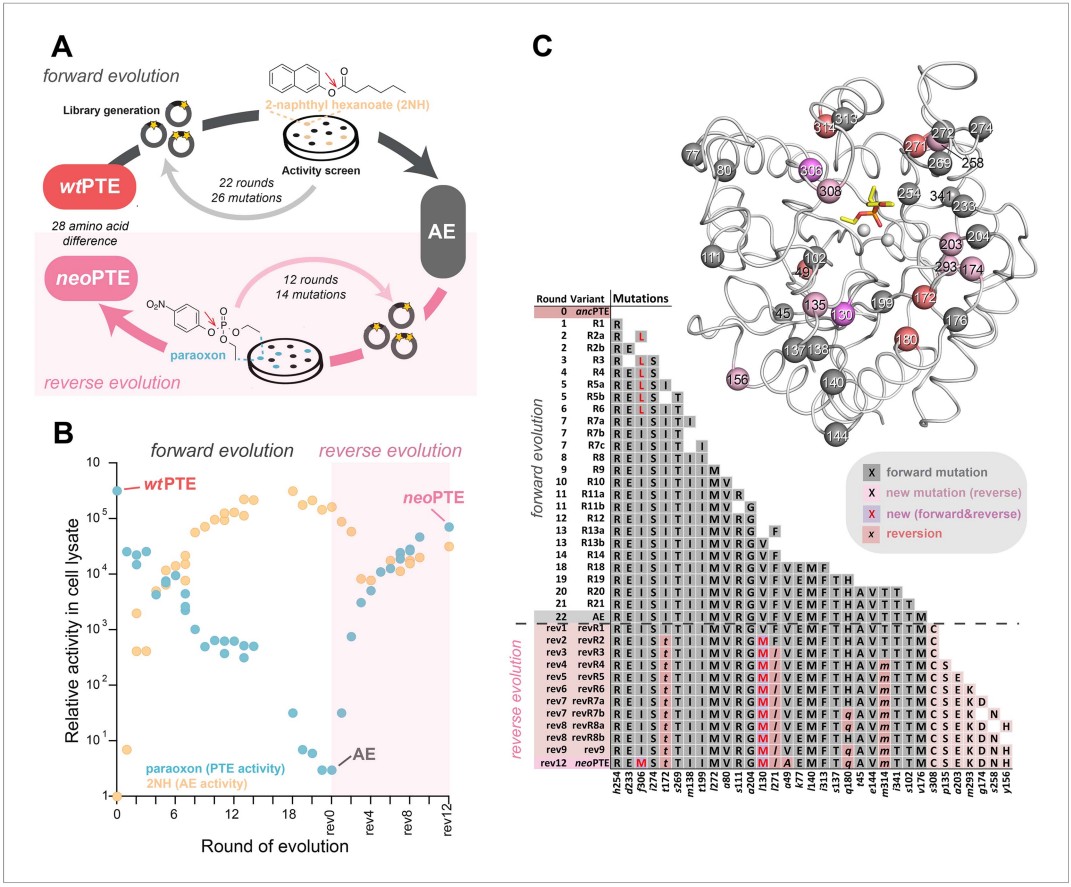

**Figure 1**. Activity and sequence changes of PTE over the evolution. (**A**) Overview of the experimental evolution. Libraries were generated and transformed into *Escherichia coli*. Proteins were expressed and screened for paraoxon and/or 2NH hydrolysis in bacterial lysates. Several thousand variants were screened per round, theoretically covering most single point mutations in the ~1000 bp PTE gene. Details are given in *Supplementary file 1*. (**B**) Activity changes during the forward (screening for arylesterase hydrolysis) and reverse evolution (screening for re-increase in phosphotriesterase hydrolysis). Steady-state kinetic parameters for all variants are provided in *Supplementary file 2A*. (**C**) Type, position, and order of occurrence of the 33 mutations obtained in the evolution. Mutations are shown relative to *wt*PTE (GenBank accession number KJ680379) with lower case italics denoting the amino acid found in *wt*PTE. Note that *wt*PTE was obtained in previous screens for improved expression levels in *E. coli* and contains six mutations relative to the naturally occurring PTE (I106L, F132L, K185R, D208G, R319S) (*Roodveldt and Tawfik, 2005*; *Tokuriki et al., 2012*). The following mutations occurred in individual variants, but were not fixated after DNA shuffling: R7a: *a*204G, R7c: *a*102V, R19: *a*78T, *v*143A, *t*311A, revR1: *c*59Y, *s*238R, revR5: *i*176V, revR8a: *d*264E, revR8b: *i*296V. All additional variants characterized and sequenced in each round are shown in *Supplementary file 1*.

The following figure supplement is available for figure 1:

**Figure supplement 1**. Correlation between activities measured in cell lysate and using purified enzyme for all variants selected over the evolution (*Supplementary file 2*).

---

*Khersonsky and Tawfik, 2010*). In the forward evolution, trade-offs then became stronger, leading to specialization of the arylesterase. The reverse evolution, however, retained characteristics of a generalist: the large increase in phosphotriesterase activity ($>10^4$-fold) was accompanied by only a small (five-fold) reduction in arylesterase activity. A possible reason for this is that the reverse evolution is still at an early phase of the functional transition after 12 rounds (vs 22 in the forward evolution). Because we were unable to isolate any variant with further improved phosphotriesterase activity, it might be necessary to impose a negative selection pressure to specialize the enzyme. The molecular basis of substrate binding and trade-offs is described further below (see also *Figure 2* and

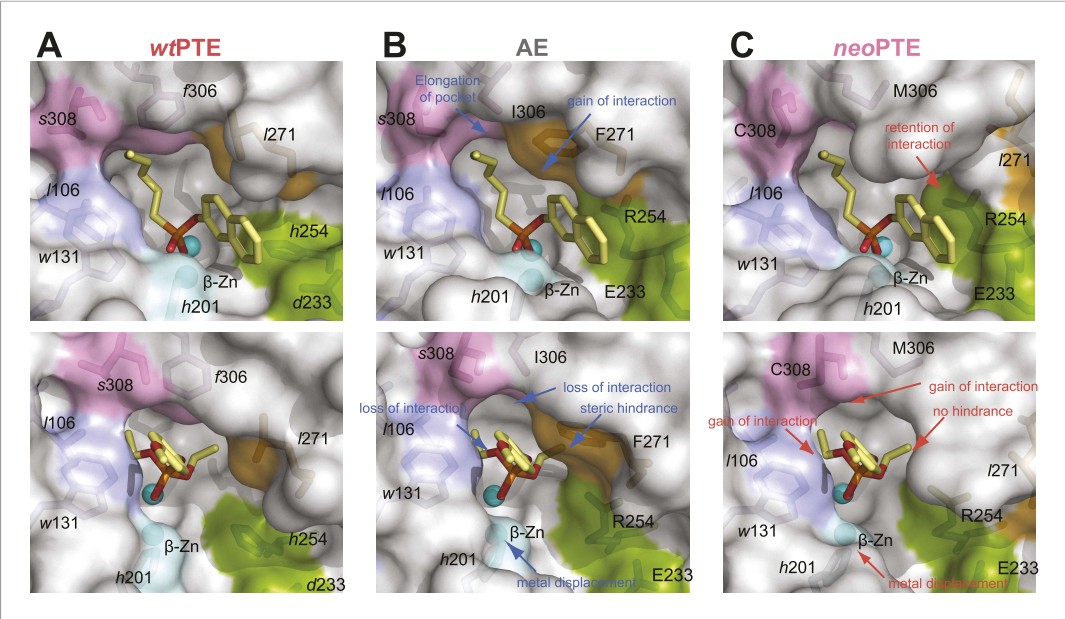

**Figure 2.** Reshaping of the PTE active site over the evolution. (**A**) *Wt*PTE (PDB ID: 4PCP) features an active site which is well adapted for paraoxon hydrolysis, but suboptimal for 2NH. (**B**) In the forward evolution, selection for arylesterase activity leads to several changes in the binding pocket from *wt*PTE to AE (PDB ID: 4PCN). (**C**) The reverse evolution leads to restoration of the ancestral state in *neo*PTE (PDB ID: 4PBF). The four regions of change are highlighted in different colors. Top row: the 2NH analogue (yellow) was modeled into the three structures by superposition with PTE-R18 in complex with the analogue (PDB ID: 4E3T) (*Tokuriki et al., 2012*). Bottom row: the paraoxon analogue diethyl 4-methoxyphenyl phosphate (yellow) was modeled into the structures by superposition with *Agrobacterium radiobacter* PTE in complex with the analogue (PDB ID: 2R1N) (*Hong and Raushel, 1996*). Amino acids found in *wt*PTE are shown in lower case italics.

The following figure supplements are available for figure 2:

**Figure supplement 1**. Details of the active site changes.

**Figure supplement 2**. Overlay of electron density maps for the active sites of (**A**) *wt*PTE (salmon) and AE (cyan) and (**B**) *wt*PTE (salmon) and *neo*PTE (magenta).

**Figure supplement 3**. Development of B-factors over the evolution.

**Figure supplement 4**. Linear free energy relationships of *wt*PTE, AE, and *neo*PTE.

*Figure 2—figure supplement 1*). Overall, we obtained a new efficient, enzyme (*neo*PTE) on par with *wt*PTE ($k_{cat}/K_M > 10^6$ $M^{-1}s^{-1}$ for paraoxon in both cases). The recovery of identical phosphotriesterase rates in *neo*PTE compared to *wt*PTE establishes that evolution of the phenotype was fully reversible.

## Genotypic irreversibility and constraints underlying phenotypic reversion

To examine the genetic changes causing phenotypic reversion, the sequence of all evolutionary intermediates was determined. Only five of the 26 mutations that accumulated in the forward evolution were reverted to the original sequence ('reversions', A49*v*, I172*t*, Q180*h*, L271*f*, M314*t*, *Figure 1C*; amino acids shown in lower case italics denote the *wt*PTE state, while amino acids not present in the wild type are shown in capital letters). Nine additional 'new mutations' accumulated, two of which occurred in positions that were mutated in the forward evolution (V130M—originally *leu*, I306M—originally *phe*), and seven were in positions that were not previously mutated (*p*135S, *y*156H, *g*174D, *a*203E, *m*293K, *s*258N, *s*308C). Overall, *neo*PTE is separated further from *wt*PTE (28 out of

333 amino acids) than AE from *wt*PTE (26 amino acids). Additional rounds of evolution failed to yield more reversions or activity increases (*Supplementary file 1*). In the forward evolution, the loss of phosphotriesterase activity was largely a side product of the property under selection, the increase in arylesterase activity. Therefore, not all mutations decreased phosphotriesterase activity (*Figure 1B*), and it is not surprising that phenotypic reversion did not require full genotypic reversion. However, a number of mutations that did contribute to decreasing phosphotriesterase in the forward process were also not reverted. Moreover, the new mutations are located in the same mutational clusters seen in the forward evolution (*Figure 1C*), indicating they may be alternative solutions to the same functional requirement and replace reversions, as detailed further below. Taken together, although the phenotype was reversible, PTE evolution was genotypically irreversible, but an alternative trajectory was readily taken.

## The active site converged towards its original state in the reverse evolution

To unravel the molecular basis of the observed genotypic irreversibility, we solved crystal structures of *wt*PTE, AE, and *neo*PTE (*Supplementary file 3*). We compared the structures and modeled both a paraoxon and a 2NH analogue into each active site (by superposition with structures containing these analogues [*Hong and Raushel, 1996*; *Tokuriki et al., 2012*]). The phosphotriester paraoxon is characterized by tetrahedral ground-state geometry and P–O cleavage proceeds via a trigonal bipyramidal transition state. The arylester 2NH is planar and C–O bond hydrolysis proceeds via a tetrahedral transition state. The structural comparison indicates that AE adapted to the planar substrate 2NH in the forward evolution, but that this came at a cost of phosphotriesterase activity, as the bulky paraoxon is no longer efficiently recognized (*Figure 2*). We identify several regions of the active site that may be responsible for the functional transition (*Figure 2* and *Figure 2—figure supplement 1*). First, a binding pocket for the naphthyl leaving group of 2NH was excavated through the combined action of *h*254R and *d*233E (*Figure 2A,B*, green region) (*Hong and Raushel, 1996*; *Tokuriki et al., 2012*). Leaving group coordination was further improved through a subtle ~1.0 Å shift of *Trp*131 (*Figure 2A,B*, purple region). Moreover, the pocket was elongated through the *f*306I mutation (*Figure 2A,B*, pink region) and narrowed by *l*271F (*Figure 2A,B*, orange region), resulting in better accommodation of the long hexanoate chain of 2NH. These changes may lead to the reduction of phosphotriesterase activity through loss of interactions (either shape complementarity, hydrophobicity, or π-π stacking) in several regions and steric hindrance in others, as described in further detail below (*Figure 2—figure supplement 1*). Additionally, the distance between the two active site zinc ions decreased from 3.8 Å to 3.3 Å (*Figure 2A,B*, light blue region and *Figure 2—figure supplement 1*). The observed structural changes are subtle, at the sub-angstrom scale, and their contributions to catalysis unquantified. However, the dispersion precision indicator (DPI; *Cruickshank, 1999*) for each of the structures is less than one-tenth of an angstrom, meaning that the observed distance changes (including the 0.5 Å shift in the metal position) are significant (*Figure 2—figure supplement 2*).

In *neo*PTE, the part of the active site necessary for phosphotriesterase activity has converged back towards its original state. The regions of suboptimal binding were re-optimized for paraoxon and the metal distance was restored to the 3.8 Å (*Figure 2C*). Moreover, the pattern of loop flexibility that is characteristic of *wt*PTE was also restored in *neo*PTE (*Figure 2—figure supplement 3*). Furthermore, we measured linear free energy relationships for *wt*PTE, AE, and *neo*PTE for both arylester and phosphotriester hydrolysis (*Figure 2—figure supplement 4*), that is, the dependence of the catalytic parameters $k_{cat}/K_M$ on the pKa of the leaving group. For phosphotriester hydrolysis by *wt*PTE, a break in pKa dependence around 7 is consistent with the rate-limiting step changing on either side of this break (*Hong and Raushel, 1996*; *Tokuriki et al., 2012*). By contrast, AE shows a continuous, linear dependence over the whole pKa range, indicating that the rate-limiting step does not change. In *neo*PTE, the pattern characteristic for *wt*PTE was restored. Together with the observed structural convergence, the simplest assumption must be that the very similar active site environment enables similar residue contributions to catalysis in *wt*- and *neo*PTE on phosphotriesterase activity. However, active site convergence is not complete, as the naphthyl binding pocket remains intact (*Figure 2C*, green region, Arg254 and Glu233), which likely explains why *neo*PTE is still bifunctional.

It should be pointed out that, at this stage, we do not know the extent to which the modification of each structural element contributes to the overall >$10^4$-fold activity change. Also, we cannot exclude the existence of alternative substrate binding modes from our model, as well as the role protein dynamics play in the functional switch. However, in the combined forward and reverse evolution, which involved a change in catalytic activity of >$10^4$ M$^{-1}$s$^{-1}$ in each direction, only four mutations were located in the active site. Instead, most functional mutations occur in more remote positions. Therefore, it is likely that fine-tuning of the active site by these remote mutations contributes significantly to the activity changes. Taken together, the restoration of all structural elements key for phosphotriesterase activity as well as the catalytic mechanism occurred despite the alternative genotypic trajectory, suggesting that biophysical requirements exist for this particular active site shape, and that phosphotriesterase activity may otherwise be inefficient.

To further investigate whether mutational accessibility is dictated by the necessity for structural convergence to the wild-type active site, a parallel evolutionary experiment was performed. In this experiment, we attempted to restore phosphotriesterase activity by a trajectory containing only new mutations. To this end, we sequenced the improved variants after each round and removed all those containing reversions. This trajectory only resulted in a 70-fold improvement in five rounds (*Figure 3A*), after which the activity plateaued and no further improved variants could be found. This failure to reach wild-type activity levels without reversions, as well as the fact that three out of the five new mutations obtained (*p*135S, *a*203E, *s*308C, *Figure 3B*) were identical to the successful trajectory containing reversions, emphasizes that the number of adaptive trajectories that lead to a wild-type level fitness peak from AE are highly limited. However, trajectories involving neutral mutations, or trajectories which do not pass through the best variant in each round but through less improved intermediates, may exist. It is likely that a wild-type-like paraoxon binding pocket is compulsory to achieve efficient phosphotriesterase activity, and only a small set of mutations (e.g., reversions or the combination of reversions and new mutations that we identified) can provide such a solution.

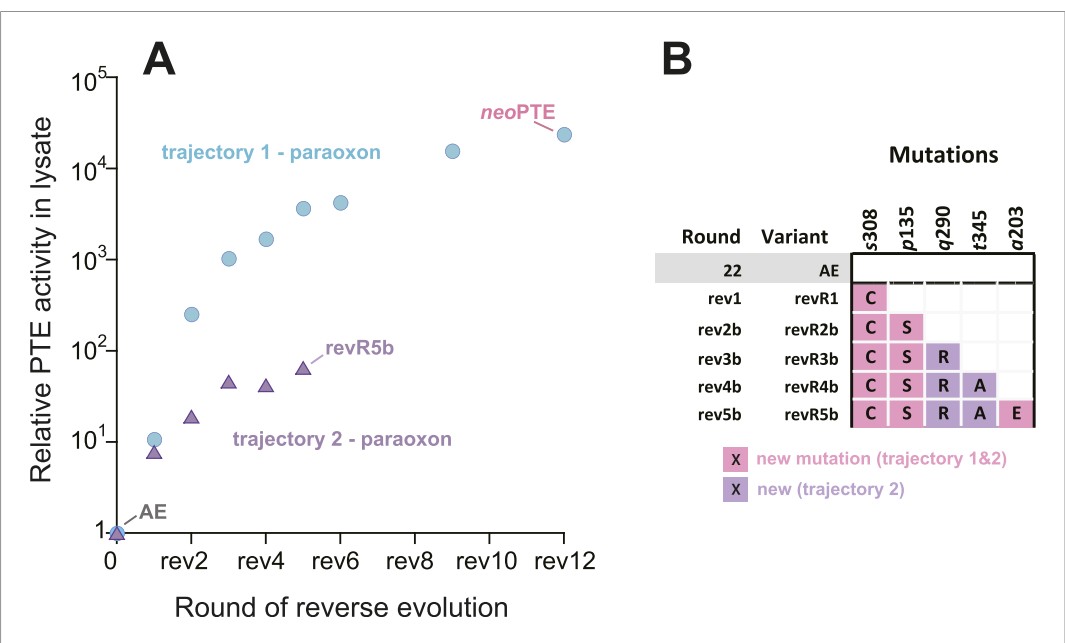

**Figure 3**. An alternative experimental evolution, where fixation of back-to-wild-type reversions was prohibited, failed to restore the original level of PTE activity. (**A**) Activity changes in the alternative trajectory. After five rounds, PTE activity plateaued at a 65-fold improvement (trajectory 2), 340-fold lower than the main trajectory (trajectory 1). (**B**) Mutations accumulated in the alternative trajectory. All clones containing reversions, which occurred frequently, were removed after sequencing and thereby prohibited from fixating. Three of the five new mutations also fixated in the main trajectory. The mutations *c*59Y and *s*238R occurred in variants revR1 and revR2b, but were not fixated after DNA shuffling. Amino acids found in *wt*PTE are shown in lower case italics. All additional variants characterized and sequenced in each round are shown in *Supplementary file 1*.

## Emergence of sequence incompatibility between the two PTEs

Next, we set out to answer the question how the two enzymes exhibiting identical phosphotriesterase activity, *wt*- and *neo*PTE, are connected on the adaptive landscape. If they populate the same fitness plateau, amino acid exchanges between them should be neutral. On the other hand, a loss of function upon interconversion between amino acids would indicate genotypic incompatibility (*Kondrashov et al., 2002*; *Lunzer et al., 2010*; *Wellner et al., 2013*), meaning that the two enzymes occupy distinct positions on the landscape that are poorly connected through a neutral network. To this end, we characterized the effect of all 28 single point exchanges separating the two enzymes in each background (56 mutants in total, *Figure 4* and *Supplementary file 2*). Mutations are considered non-neutral if they cause a >1.3-fold change in phosphotriesterase activity in lysate compared to the parent background because, in our screening system, this cut-off enabled us to reliably identify improved variants. Moreover, we have performed a statistical analysis of the mutational effects, which confirms that a >1.3-fold change is significant (p-values <0.05) in almost all cases (statistics are provided in *Figure 4—source data 1* and *Supplementary file 2B*). According to this analysis, only eight of 28 exchanges were compatible; they were neutral in both backgrounds (*Figure 4A*). The remaining 20 positions showed incompatibility, 15 of which were partially incompatible, as the exchange was neutral in one background but deleterious in the other (*Figure 4B,C*). Five exchanges were completely incompatible; they severely decreased activity in both backgrounds (*Figure 4D*). Taken together, despite >90% sequence identity between *wt*- and *neo*PTE, the reverse trajectory led to a functional sequence that is poorly connected with the original one. It remains unknown whether the two sequences comprise completely separate peaks on the adaptive landscape or are connected through a neutral network, that is, if the neutral exchanges would permit the subsequent occurrence of initially deleterious exchanges. However, because >70% of the mutated positions cause incompatibility, only one out of the 54 exchanges confers higher fitness, and this exchange (*neo*PTE + *f*306M) would require two simultaneous base changes, it is unlikely that an evolutionary transition between the two could easily occur by adaptive or strong purifying selection.

## Mutational epistasis underlies genotypic irreversibility

To understand how convergence to the original function and architecture was achieved despite genotypic irreversibility and incompatibility, we performed a comprehensive mutational analysis. All 33 mutated positions were examined in the background of the three enzymes (*wt*PTE, AE, and *neo*PTE), and in the background in which they originally occurred in the evolution (i.e., in the different rounds) to identify mutations that are epistatic—that is, change their effect depending on the genetic background (*Figure 5* and *Supplementary file 2*). To determine whether or not the measured changes were significant, the same stringent cut-off as described above for the comparison between *wt*- and *neo*PTE was applied (statistics are provided in *Supplementary file 2B–G*). Furthermore, we analyzed the crystal structures to determine which mutations caused the divergence and convergence of the active site configuration.

The analysis revealed extensive epistasis during the forward and reverse evolution. In the forward evolution, the effect of mutations is significantly altered after their fixation due to epistasis caused by mutations subsequently accumulated in the trajectory. For example, some mutations initially increased (*t*172I in round 6 and *l*271F in round 14) or were neutral to (*l*130V in round 14) phosphotriesterase activity when they occurred in the trajectory, and were thus unfavorable to revert as their reversion would not change (V130*l*) or decrease (I172*t*, F271*l*) activity (*Figure 6A*). However, reversion of these mutations became possible (i.e., would lead to an increase in activity) in the background of AE (*Figure 6A*). On the contrary, *h*254R decreased phosphotriesterase activity when it occurred in round 1 and therefore its reversion (R254*h*) would initially be favorable. However, the effect of this reversion switched to unfavorable (R254*h*) when it was tested in AE (*Figure 6B*). Moreover, mutations in the forward evolution had a permissive effect on the accumulation of new mutations and, in this way, opened up a path towards the alternative trajectory taken in the reverse process; all new mutations had a neutral or negative effect on phosphotriesterase activity in the genetic background of *wt*PTE but most of them become positive in AE (*Figure 6C*); for example, AE-s308C (6.4-fold), AE-V130M (5.4-fold) and AE-*p*135S (2.9-fold). Because these mutations can compete with the most favorable reversions (>1.3–8-fold effect, *Supplementary file 2*), they were selected in the early rounds of the reverse evolution, laying the foundation for the alternative trajectory.

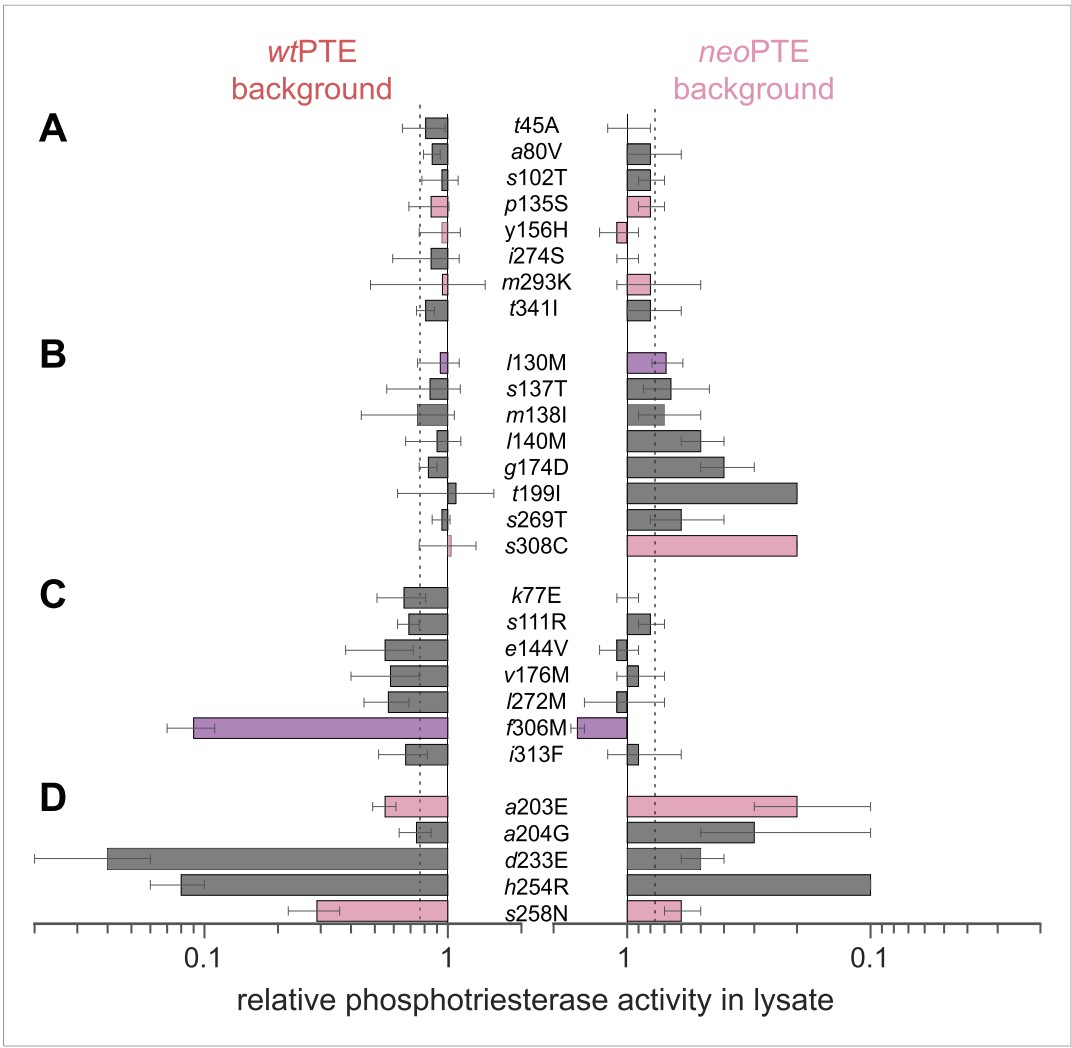

**Figure 4**. Genotypic incompatibility between *wt*PTE and *neo*PTE. (**A–D**) The effect of the 28 amino acid exchanges separating the two enzymes was tested in the background of *wt*PTE and *neo*PTE, respectively. Activities are given relative to the parent mutational background, *wt*PTE or *neo*PTE. Amino acids found in *wt*PTE are shown in lower case italics. Color code as in *Figure 1*. (**A**) Compatible exchanges, neutral in both backgrounds. (**B**, **C**) Partially incompatible exchanges, neutral in one background but deleterious for another. (**D**) Mutually incompatible exchanges. Mutations causing a >1.3-fold change compared to the respective parent mutant (dotted line) are considered non-neutral. p-values compared to each parent (*Supplementary file 2B*) and p-values for the effect of each mutation *Figure 4—source data 1* in the two backgrounds were calculated. Note that the effect of *i*313F, which causes a significant decrease in *wt*PTE, is statistically not significant between *wt*PTE and *neo*PTE.

The following source data is available for figure 4:

**Source data 1**. Comparison of the effect of mutations in *wt*- and *neo*PTE.

In the reverse evolution, the active site architecture necessary for phosphotriesterase activity was restored largely through new mutations, which restricted the reversion of mutations accumulated in the forward evolution. Overall, nine of the 10 reversions that were initially favorable in the background of AE lost their favorable effect in *neo*PTE because of epistasis during the reverse evolution (*Figure 7A*). We were able to trace the molecular basis of this effect in several cases as described in the following. First, *f*306I enlarged the active site in the forward evolution, resulting in a loss of shape complementarity to paraoxon. In the reverse evolution, the nearby *s*308C offsets this effect by increasing the hydrophobicity of the pocket (*Figure 2*, *Figure 7C*, *Figure 2—figure supplement 1A*).

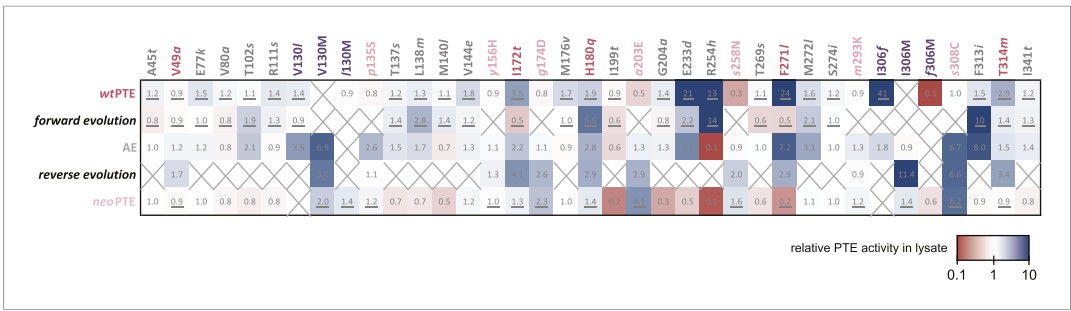

**Figure 5**. Changes in phosphotriesterase activity upon mutations in five different backgrounds: *wt*PTE in the forward evolution, AE in the reverse evolution, and *neo*PTE. Thirty-three positions were mutated in the entire evolution, two of which (130 and 306) were mutated to two different amino acids. Amino acids found in *wt*PTE are shown in lower case italics. Numbers indicate the fold change in activity caused by a mutation in a certain background (**Supplementary file 2B–F**). Mutations causing a >1.3-fold change compared to the respective parent mutant are considered non-neutral. p-values compared to each parent were calculated (**Supplementary file 2B,D,F**). The mutations T341*i* in AE, *l*140M and *t*199I in the forward evolution, and V49*a* and *s*258N in the reverse evolution are not significant (p-values >0.05). Therefore, out of 144 mutations, only five show a >1.3-fold effect, but are statistically not significant. Boxes that are crossed out indicate that a mutation did not occur in this background. For direct comparison, the activity changes resulting from a mutation are adjusted to the same direction—from the amino acid found in AE to the respective other amino acid (label at the top). To illustrate, the effect of R254*h* was measured as follows: AE and *neo*PTE contain Arg254, and thus the effect of R254*h* is directly calculated based on the comparison between AE and AE-R254*h* (Fold change$_{R254h}$ = Acticity$_{AE-R254h}$/Activity$_{AE}$) and between *neo*PTE and *neo*PTE-R254*h* (Fold change$_{R254h}$ = Acticity$_{neoPTE-R254h}$/Activity$_{neoPTE}$). However, because *wt*PTE and the forward evolution background already contain *His*254, the effect of introducing this amino acid has to be calculated 'in reverse' by first assuming to remove this mutation and then adding it back in, that is, based on the comparison between *wt*PTE-*h*254R and *wt*PTE (Fold change$_{R254h}$ = Acticity$_{wtPTE}$/Activity$_{wtPTE-h254R}$). All mutational effects that were calculated in this 'reverse' way are underlined. Note that *wt*PTE-*h*254R is identical to the round 1 variant and therefore the effect in the forward evolution is the same as in the *wt*PTE background. Because R254*h* did not occur in the reverse evolution, no effect could be calculated in this background and the respective box is crossed out.

The redundancy of the mutations *f*306I and *s*308C was also evidenced by combinatorial mutational analysis; incorporation of *s*308C restricts subsequent reversion of I306*f* due to sign epistasis (**Figure 7B**). While this reversion would have been favorable in isolation, phosphotriesterase activity of the double mutant AE-I306*f*-*s*308C is reduced compared to AE-*s*308C. Second, the active site was narrowed in the forward evolution by *l*271F and several other mutations in loop7/8 including *l*272M and *i*313F (**Figure 7C** and **Figure 2—figure supplement 1B**), causing steric hindrance for paraoxon. The pocket was re-opened initially by the reversion F271*l*. Subsequently, the new mutation *s*258N destabilized and altered the conformation of loop 7 and further enlarged the pocket (**Figure 2**, **Figure 2—figure supplement 1B**). We also observed that incorporation of *s*308C and F271*l* restricted the reversion of both *l*272M and *f*313I (M272*l* and I313*f*, **Figure 7B**). Third, the active site was reshaped by a subtle ∼1.0 Å shift of *Leu*106 and *Trp*131, which was likely triggered by a cluster of remote mutations occurring in the same loops (*s*102T, *l*130V, *m*138I, *s*137T, and *v*140M, **Figure 7D**, and **Figure 2—figure supplement 1C**). In the reverse evolution, these residues are shifted back to their original positions through two new remote mutations, *p*135S and V130M (**Figure 7D**). Again, the two mutations are redundant and mutually exclusive; *p*135S restricts the reversion of *m*138I (I138*m*, **Figure 7B**). Fourth, the distance between the two active site zinc ions decreased from 3.8 to 3.3 Å in the forward evolution through displacement of the metal-chelating *His*201 and the β-metal (**Figure 7E**), which was likely triggered by the combined action of several remote mutations in loops 4 and 5 (*t*172I, *q*180H, *t*199I, and *a*204G, **Figure 2**). In the reverse evolution, the positions of *His*201 and the β-metal, as well as the original inter-metal distance of 3.8 Å, were restored through the reversion I172*t* and formation of a new hydrogen bonding network with two additional new mutations, *a*203E and *g*174D (**Figure 7E**). These examples demonstrate that rewiring the intramolecular interaction network of the protein can result in the same physical solution in key elements in the active site. Rewiring occurs because new mutations act as 'epistatic ratchets'

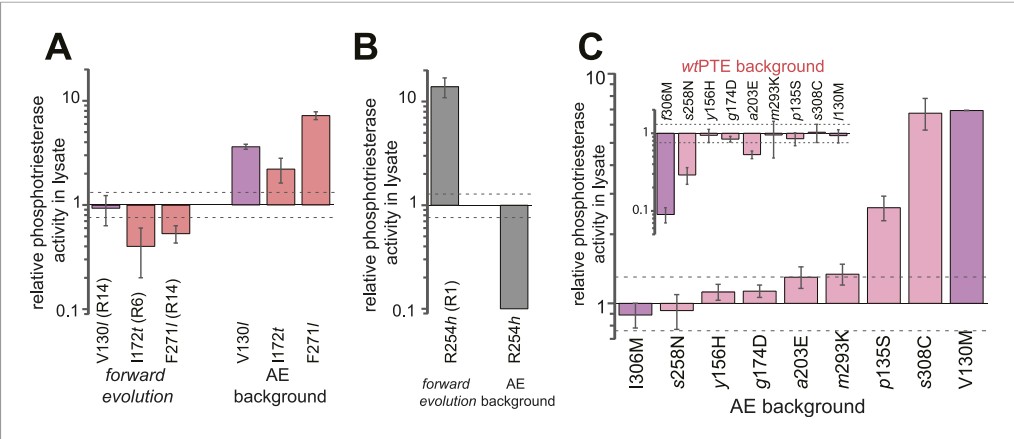

**Figure 6**. Epistasis between mutations in the forward evolution restricts some reversions while permitting others as well as new mutations. (**A**) Several reversions change their effect from unfavorable upon their initial occurrence in the forward evolution to favorable in AE. (**B**) Other reversions change their effect from favorable to unfavorable. Note that, in the forward evolution, mutations occurred in the opposite direction as shown (*l*130V, *t*172I, *l*271F, and *h*254R), but are given in the same direction as AE for direct comparison. Phosphotriesterase activity was too low to be determined in AE + R254h, but at least 10-fold reduced. (**C**) The effect of new mutations changes from *wt*PTE (small panel) to AE (large panel). Relative activities were calculated by comparing a variant containing a certain mutation with one lacking only this mutation. Mutations causing a >1.3-fold change compared to the respective parent mutant (dotted line) are considered non-neutral. p-values compared to each parent (***Supplementary file 2B***) and p-values for the effect of each mutation in the two respective backgrounds shown in each panel were calculated (***Figure 6—source data 1, 2***). Note that the mutation *m*293K, which causes a significant increase in AE, does not have a significantly different effect in the two backgrounds. Amino acids found in *wt*PTE are shown in lower case italics. Color code as in ***Figure 1***. All other mutational effects in the different backgrounds are given in ***Figure 5*** and ***Supplementary file 2***.

The following source data are available for figure 6:

**Source data 1**. Comparison of the effect of mutations in the forward evolution and in AE (panels A, B).

**Source data 2**. Comparison of the effect of mutations in *wt*PTE and AE (panel C).

(***Bridgham et al., 2009***) for potential reversions, restricting their fixation and thus leading to the incompatible new enzyme *neo*PTE.

## Discussion

Our work demonstrated that a >$10^4$-fold loss in phosphotriesterase activity, which accompanied the functional transition to a distinct chemical reaction—arylester hydrolysis—via accumulation of 26 mutations, is readily restored when the selection pressure is reverted. Phenotypic reversal has been observed in previous cases (***Clarke, 1985***; ***Lenski, 1988***; ***Crill et al., 2000***; ***Teotonio and Rose, 2000***; ***Kitano et al., 2008***), supporting the notion that the phenotype is largely subject to deterministic forces. The likelihood of evolutionary reversibility depends on the complexity of the system and the distance in sequence and function from the ancestor, and it is possible that starting from a more distantly evolved arylesterase would have failed to restore phosphotriesterase function. Moreover, we modulated protein stability throughout the entire trajectory using overexpression of GroEL/ES to avoid evolutionary dead ends caused by stability bottlenecks (***Socha and Tokuriki, 2013***; ***Wyganowski et al., 2013***). In the absence of chaperones, adaptation may have occurred through a different pathway. Another limitation of our work is that we only examined two evolutionary trajectories (the main trajectory and the trajectory without reversions). One could imagine conducting multiple parallel evolutionary experiments to shed light on the repeatability of the trajectory taken, but unfortunately our screening system is not amenable to such a throughput. Nevertheless, our experiment shows that the genotype is subject to strong constraints: an alternative mutational

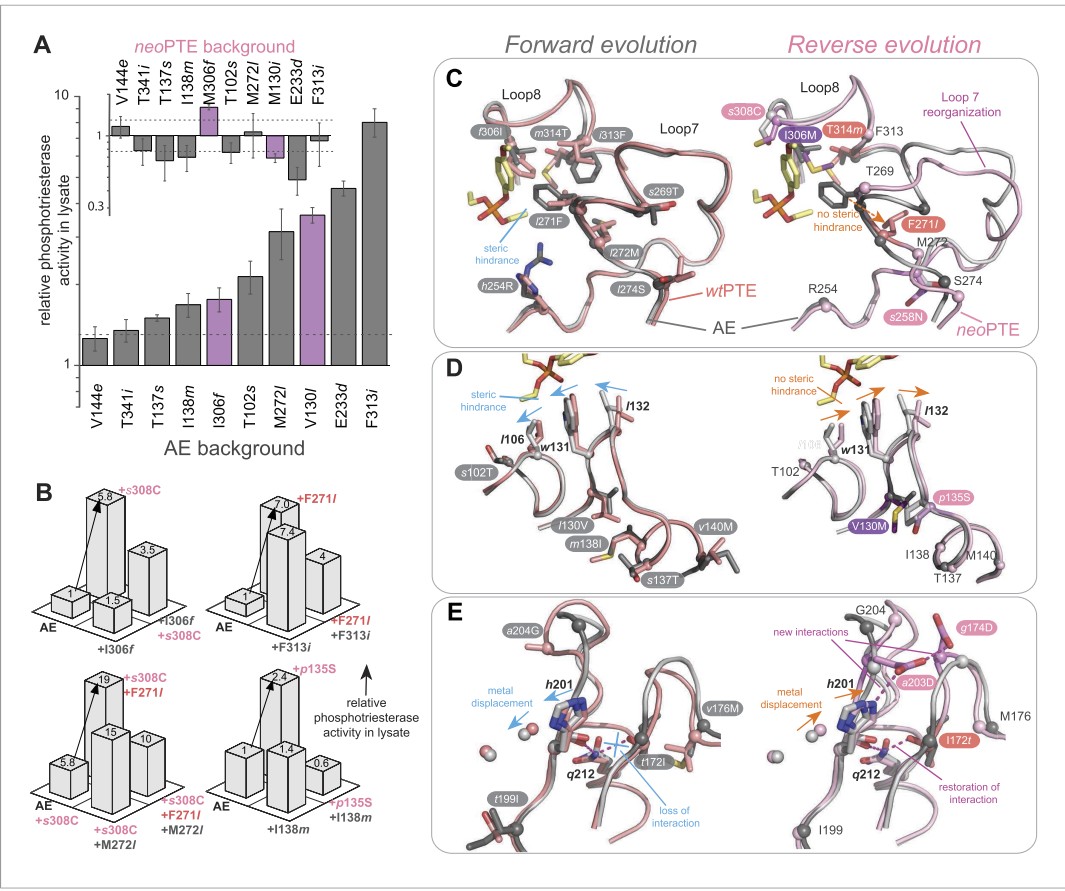

**Figure 7**. Convergence to the original active site configuration in the reverse evolution through rewiring of the molecular interaction network leads to genetic incompatibility. (**A**, **B**) Epistasis during the reverse evolution causes irreversibility and incompatibility. (**A**) The activity change of mutations that were favorable in the initial stage of reverse evolution, but not reverted. *neo*PTE background: small panel; AE background: large panel. Color code as in *Figures 1, 2*. Mutations causing a >1.3-fold change compared to the respective parent mutant (dotted line) are considered non-neutral. p-values compared to each parent (*Supplementary file 2B*) and p-values for the effect of each mutation in the two respective backgrounds shown in each panel were calculated (*Figure 7—source data 1*). (**B**) Combinations of mutations that constrained the evolutionary trajectory due to sign epistasis. Phosphotriesterase activity is shown on a linear scale. p-values are given in *Supplementary file 2G*. Note that the two mutants AE + F271*l* + *s*308C and AE + M272*l* + *s*308C have non-significant p-values compared to the 'double mutant' in this series, AE + F271*l* + M272*l* + *s*308C. However, determination of $k_{cat}/K_M$ values confirms sign epistasis in this series (see also *Supplementary file 2G*). (**C–E**) Amino acid changes in the forward (left panel) and reverse evolution (right panel). (**C**) Reorganization of loops 7 and 8. A new mutation, *s*258N, caused the reorganization (see also *Figure 2—figure supplement 1A,B*). (**D**) Different combinations of remote mutations in loop 3 resulted in identical positioning of *Leu*106, *Trp*131, and *Leu*132 in *wt*PTE and *neo*PTE (see also *Figure 2—figure supplement 1C*). (**E**) Rewiring the interaction network in *neo*PTE by remote mutations in loops 4 and 5 led to β-metal displacement (see also *Figure 2—figure supplement 1D*). Amino acids found in *wt*PTE are shown in lower case italics.

The following source data is available for figure 7:

**Source data 1**. Comparison of the effect of mutations in *wt*PTE and AE (panel A).

pathway was taken which prevented retracing of the original pathway. Genotypic irreversibility was caused by several factors. First, because selection in the forward evolution was only for increased arylesterase activity (except in rounds 19–22), the effect of the mutations on phosphotriesterase activity was stochastic: most decreased phosphotriesterase activity, some did not affect it, and some increased phosphotriesterase activity. Therefore, even if one were to revert the mutations in the reverse order of their occurrence (from rounds 22 back to 1), the lack of continuous activity increases

would prevent a gradual adaptive trajectory. Second, by the end of the forward evolution, several new mutations able to increase paraoxonase activity emerged due to epistasis. The fixation of these mutations then acts as an epistatic ratchet (*Bridgham et al., 2009*) that prevents reversions. Therefore, as soon as the first new mutation accumulates, the trajectory deviates further from the original path.

Our work suggests that only certain sets of mutations are able to cause phenotypic reversal. Although genotypic redundancy was observed, the presence of at least some reversions was essential for complete restoration of catalytic activity, and several new mutations were shared between the two trajectories examined. Similarly, other experimental evolution studies that examined parallel evolutionary trajectories starting from the same sequence often resulted in accumulation of the same mutations (*Bull et al., 1997*; *Salverda et al., 2011*; *Dickinson et al., 2013*; *Khanal et al., 2014*). These observations indicate that a number of functional mutations accessible from a particular starting point are highly limited, and that the genotype is also subjected to deterministic forces to some extent. In our case, the limited accessibility to functional mutation can be explained by the requirement to adapt the wild-type active site configuration in order to obtain efficient phosphotriesterase hydrolysis. Recent work by Harms et al. showed that the accessibility of functional and permissive mutations on hormone receptors is also strongly constrained by biophysical requirements imposed on the binding pocket as well as by protein dynamics (*Harms and Thornton, 2014*). Understanding such biophysical requirements and, in the case of enzymes, imperatives of chemical reactivity, is essential to develop our knowledge of evolutionary dynamics and constraints, although the exact nature of such requirements may be unique to each protein.

In the case of PTE, the combination of multiple subtle changes is required to fulfill these biophysical requirements and completely switch the enzyme's ability to recognize two different substrates (paraoxon vs. 2NH: tetrahedral vs. planar, P–O bond vs C–O bond cleavage, trigonal bipyramidal vs. tetrahedral transition state geometry). All but four of the 33 mutations occur in locations remote from the active site and act by fine-tuning rather than directly changing the active site configuration. Some changes occur at the sub-Å level (e.g., the shift in *Trp*131, *Leu*132, and the β-metal), and possibly act by influencing the dynamics of the active site loops. It may be that only remote mutations can achieve such subtle optimization. A mutation directly in the active site would result in a larger, more disruptive change (e.g., even a single additional carbon center would fill an additional 4 Å radius) and therefore be unable to provide the necessary fine-tuning. Other directed evolution studies also observed the accumulation of remote mutations (*Morley and Kazlauskas, 2005*), suggesting that fine-tuning of the active site may be a common strategy to implement a new function.

Our work reveals that the adaptive landscape of PTE is highly rugged: even single amino acid changes can regulate activities upwards or downwards and also predetermine the potential effect of subsequent mutations. As discussed above, because multiple mutations can directly or indirectly affect the same key component for catalysis, their effects are likely to be epistatic. Therefore, the alternative trajectory is caused by epistasis between mutations: frequently, those mutations that accumulate first have a permissive or restrictive effect on subsequent mutations. Overall, >70% of mutations have highly variable effects on phosphotriesterase activity, depending on the genetic background (26 out of 33 positions, *Figure 5*), and ~40% showed sign epistasis (7 out of 33). The role of epistasis in natural evolution has recently received much attention, but its extent and prevalence are still under debate (*Whitlock et al., 1995*; *Poelwijk et al., 2007*; *de Visser et al., 2011*; *Breen et al., 2012*; *Harms and Thornton, 2013*; *McCandlish et al., 2013*; *Kaltenbach and Tokuriki, 2014*). Our findings suggest a high frequency of strong epistatic interactions during functional adaptation and therefore support the view that epistasis is paramount in shaping evolution. However, while restrictive mutations block many of the possible evolutionary trajectories, as has been previously emphasized, permissive mutations simultaneously open up new pathways, avoiding 'evolutionary dead-ends' and contributing to the diversity of enzyme homologs found in nature.

Moreover, our study demonstrates how genotypic irreversibility leads to the emergence of a functional sequence incompatible with the original one (*Kondrashov et al., 2002*; *Maheshwari and Barbash, 2011*), and implies the importance of evolutionary contingency on the genotypic level. In nature, the environment never ceases to change and a temporary relaxation in selection pressure (i.e., the level and type of nutrients or toxins) followed by re-adaptation (through both reversions and new compensatory mutations) may be common (*Akashi et al., 2012*). Higher levels of organization (such as metabolic or regulatory networks) might be subject to similar contingency; restoration of

a certain function may be achieved by alternative mutations in other parts of the protein structure, in other domains, or in a different protein altogether. If the mutations are mutually exclusive, sequence incompatibilities may arise rapidly. Therefore, in addition to proposed mechanisms such as genetic drift (*Akashi et al., 2012*), genotypic irreversibility may contribute to the prevalence of incompatibility between orthologous enzymes (*Lunzer et al., 2010*; *Kvitek and Sherlock, 2011*; *Corbett-Detig et al., 2013*; *Wellner et al., 2013*; *Schumer et al., 2014*; *Shafee et al., 2015*).

Finally, our observations have important implications for the engineering of highly efficient enzymes—for example, how fine-tuning of multiple active site regions can confer significant activity changes, and how context-dependent such changes are. As our understanding of protein sequence-structure-function relationships grows, further rational and computational approaches need to be developed to address the role of remote mutations and epistasis to enhance our ability to create tailor-made proteins.

## Materials and methods

### Error-prone PCR

Error-prone PCR libraries were generated using nucleotide analogues (8-oxo-2′-deoxyguanosine-5′-triphosphate [8-oxo-dGTP] and 2′-deoxy-P-nucleoside-5′-triphosphate [dPTP]) or Mutazyme (GeneMorph II Random Mutagenesis kit, Agilent, Santa Clara, CA, United States). A typical protocol using nucleotide analogues can be found in *Tokuriki and Tawfik (2009)*. A typical protocol using Mutazyme starts with a 50 µl PCR reaction containing 50 ng of pET-Strep-PTE template and 0.8 µM of outer primers (forward TTCCCCATCGGTGATGTC, reverse GTCACGCTGCGCGTAAC). Cycling conditions were: initial denaturation at 95°C for 2 min followed by 10 cycles of denaturation (30 s, 95°C), annealing (30 s, 63°C) and extension (1 min, 72°C), and a final extension step at 72°C for 10 min. Plasmid was removed by treatment with *Dpn* I (NEB, Ipswich, MA, United States). The PCR product was purified using the QIAquick PCR purification kit (Qiagen, Netherlands), and amplified further with BIOTAQ DNA polymerase (Bioline, United Kingdom) using inner primers (forward ACGATGCGTCCGGCGTA, reverse GCTAGTTATTGCT CAGCG) and starting from 20 ng of template in a 100 µl reaction volume. Cycling conditions were: initial denaturation at 95°C for 2 min followed by 20 cycles of denaturation (30 s, 95°C), annealing (1 min, 58°C) and extension (30 s, 72°C), and a final extension step at 72°C for 2 min. This gave an average of two amino acid substitutions per gene.

### DNA shuffling

PTE genes of selected variants were amplified by PCR from pET-Strep-PTE plasmids using the outer primers and BIOTAQ DNA polymerase. Cycling conditions were: initial denaturation at 95°C for 2 min followed by 25 cycles of denaturation (30 s, 95°C), annealing (1 min, 63°C) and extension (1 min, 72°C), and a final extension step at 72°C for 2 min. PCR products were purified using the QIAquick PCR purification kit and mixed at equal amounts. Before the preparative digest, conditions were optimized by digesting 1 µg of template DNA with a range of *DNase* I concentrations (Fermentas, Waltham, MA, United States). DNase digest buffer (10×) consists of 0.5 M Tris-HCl pH 7.5 supplemented with 0.5 mg/ml BSA. In addition, reactions contained 10 mM MnCl₂. Reactions were incubated for 10 min at 37°C, stopped by addition of 1/5 vol of stop buffer (30 mM EDTA pH 8.0, 30% glycerol and ≈0.6× of a DNA loading buffer) and analyzed by agarose gel electrophoresis in TBE buffer (2% agarose gel, 45 mM Tris, 45 mM boric acid, and 1 mM EDTA pH 8.0; for all other agarose gel electrophoresis procedures, we used 1% agarose gels and TAE buffer, which is 40 mM Tris, 20 mM acetic acid, and 1 mM EDTA pH 8.0). Reactions were scaled up to 10–15 µg of DNA and the digest repeated at the appropriate DNase dilution to give fragments in the range of 50–150 bp. Fragments were purified by gel extraction and 60–80 ng used in a 20 µl assembly PCR. This PCR was performed with Herculase I (Stratagene, La Jolla, CA, United States). Cycling conditions were: initial denaturation at 96°C for 90 s followed by 35 cycles of denaturation (30 s, 94°C), annealing (incremental 3°C steps from 65°C down to 41°C, 90 s each) and extension (2 min, 72°C), and a final extension step at 72°C for 10 min. Full-length assembly products were amplified using the inner primers and BIOTaq DNA polymerase under the following cycling conditions: initial denaturation at 95°C for 2 min followed by 25 cycles of denaturation (30 s, 95°C), annealing (1 min, 58°C) and extension (1 min, 72°C), and a final extension step at 72°C for 2 min. The amount of assembly product used as template for this reaction was varied and

product formation verified by 1% agarose gel electrophoresis. Fractions containing product were pooled and purified using the QIAquick PCR purification kit.

## Construction of single and double mutants

Mutants were constructed by site-directed mutagenesis as described in the QuikChange Site-Directed Mutagenesis manual (Agilent).

## Cloning

PCR products and pET-Strep-ACP vector were digested with Fermentas FastDigest *Nco* I and *Hind* III (or *Kpn* I, see *Supplementary file 1*) for 1 hr at 37°C. The vector was treated with CIP (calf-intestinal alkaline phosphatase, NEB, Ipswich, MA, United States) for an additional hour and subsequently insert and vector were purified from 1% agarose gel using the QIAquick gel extraction kit followed by the Qiagen PCR purification kit. Ligations were performed at a vector:insert mass ratio of 1:1 using T4 DNA ligase (NEB, Ipswich, MA, United States) supplemented with 0.5 mM ATP (NEB, Ipswich, MA, United States) for 2 hr at 22°C or 16°C overnight. Prior to transformation, reactions were purified by ethanol/glycogen (Fermentas, Waltham, MA, United States) precipitation. Transformation into electrocompetent E. cloni 10G (Lucigen, Middleton, WI, United States) yielded at least $10^5$ colonies.

## Pre-screen on agar plates

Plasmids were extracted and re-transformed into *E. coli* BL21 (DE3) containing pGro7 plasmid for overexpression of the GroEL/ES chaperone system. Transformation reactions were plated on an average of 10 agar plates (140 mm diameter) containing 100 µg/ml ampicillin (or 50 µg/ml kanamycin, see *Supplementary file 1*) and 34 µg/ml chloramphenicol such that each plate contained 200–1000 colonies, leading to a final library size of 2000–10,000 variants. Colonies were transferred onto nitrocellulose membrane (BioTrace NT Pure Nitrocellulose Transfer Membrane 0.2 µm, PALL Life Sciences, Port Washington, NY, United States), which was then placed onto a second plate additionally containing 1 mM isopropyl β-D-1-thiogalactopyranoside (IPGT) 200 µM $ZnCl_2$ (to ensure availability of $Zn^{2+}$ ions necessary for enzymatic activity), and either 20% (wt/vol) arabinose for chaperone overexpression or 20% (wt/vol) glucose for repression of chaperone expression. After expression overnight at room temperature for plates containing arabinose or for 1 hr at 37°C for plates containing glucose, the membrane was placed into an empty petri dish. For low activity levels of the parent gene where a maximum signal is desirable, cells were lysed prior to the activity assay by alternating three times between storage at −20°C and 37°C. For higher activities, the lysis step was omitted, making it easier to differentiate between different colonies. For the activity assay, 20–25 ml of 0.5% Agarose in 50 mM Tris-HCl buffer, pH 7.5 containing 200 µM 2NH (Sigma, St. Louis, MO, United States) and Fast Red (Sigma, St. Louis, MO, United States) was poured onto the membrane. Red color developed within 30 min. To screen for phosphotriesterase activity, the buffer contained varying concentrations of fluorogenic phosphotriester instead of 2NH/Fast Red as indicated in *Supplementary file 1*. Turnover of O-fluoresceinyl-O,O-diethyl-thiophosphate (fluoresceinyl-DETP, excitation 495 nm, emission 520 nm) was detected in a Typhoon 9400 scanner (GE Healthcare, Wauwatosa, WI, United States) after an appropriate incubation time (0–3 hr). In the case of 7-O-diethylphosphoryl-3-cyano-4-methyl-7-hydroxycoumarin (Me-DEPCyC), activity was detected in an agarose gel imager (excitation 365 nm) using a SYBR Safe filter.

## Screens in 96-well plates

Colonies exhibiting high enzymatic activity identified in the pre-screen were picked and re-grown in four to six 96-deep well plates overnight at 30°C, leading to a library of 400–600 pre-selected variants. Wells contained 200 µl lysogeny broth (LB) supplemented with 100 µg/ml ampicillin and 34 µg/ml chloramphenicol. Subsequently, deep well plates containing 500 µl LB per well supplemented with ampicillin, chloramphenicol, and 20% (wt/vol) arabinose or glucose (depending on whether chaperone overexpression was to be induced or repressed) were inoculated with 25 µl of pre-culture and grown for 2–3 hr at 37°C until the $OD_{600}$ reached ~0.6. Expression of PTE variants was induced by adding IPTG to a final concentration of 1 mM and cultures were incubated for an additional 2 hr at 30°C or for 1 hr at 37°C in rounds aimed at reducing chaperone dependence. Cells

were spun down at 4°C at maximum speed (3320×g) for 5–10 min and the supernatant was removed. Pellets were frozen for a minimum of 30 min at −80°C and subsequently lysed by addition of 200 μl 50 mM Tris-HCl pH 7.5 supplemented with 0.1% (wt/vol) Triton-X100, 200 μM ZnCl$_2$, 100 μg/ml lysozyme, and ~1 μl of benzonase (25 U/μl, Novagen, Madison, WI, United States) per 100 ml. After 30 min of lysis at room temperature, cell debris was spun down at 4°C at 3320×g for 20 min. Depending on the activity level of the library, clarified lysate was diluted prior to the activity assay to obtain a good signal in the initial linear phase of the reaction. Reactions were performed in transparent 96-well plates containing 200 μl per well (20 μl lysate + 180 μl of 200 μM substrate in 50 mM Tris-HCl, pH 7.5 supplemented with 0.02% Triton-X100 in the case of paraoxon and 0.1% in the case of 2NH/FR). Paraoxon hydrolysis was monitored at 405 nm; 2NH hydrolysis was monitored at 500 nm via complex formation with Fast Red. Improvements >1.3-fold relative to the previous round were considered significant. The best clones were picked and re-grown in triplicate. The observed initial rates were normalized to cell density (determined by the OD$_{600}$) and the average values determined. Approximately 10 improved variants were sequenced after each round. A description of each directed evolution round including selection criteria, the mutations found in each sequenced variant, and mention of the variants chosen as templates for the next library generation can be found in *Supplementary file 1*.

## Purification of Strep-tagged proteins for enzyme kinetics

pET-Strep-PTE plasmids were transformed into *E. coli* BL21 (DE3) and grown at 37°C in TB medium containing 100 μg/ml ampicillin and 200 μM ZnCl$_2$. Expression was induced with 0.4 mM IPTG when cell density reached an OD$_{600}$ of 0.6 units and cells grown overnight at 20°C. Cells were harvested by centrifugation at 3320×g and 4°C for 10 min, resuspended and lysed for 1 hr at room temperature using a 1:1 mixture of B-PER Protein Extraction Reagent (Thermo Scientific, Waltham, MA, United States) and 50 mM Tris-HCl buffer, pH 7.5 containing 200 μM ZnCl$_2$, 100 μg/ml lysozyme and ~1 μl of benzonase per 100 ml. Cell debris was removed by centrifugation at 30,000×g and 4°C for 45 min and the clarified lysate passed through a 45 μm filter before loading onto a Strep-Tactin Superflow High capacity column (1 ml column volume). After several washes with 50 mM Tris-HCl buffer, pH 7.5 containing 200 μM ZnCl$_2$, Strep-PTE variants were eluted in the same buffer containing 2.5 mM desthiobiotin according to the manufacturer's instructions (IBA BioTAGnology, Germany). Protein was dialyzed overnight against 50 mM Tris-HCl buffer, pH 7.5 containing 100 mM NaCl and concentrated if necessary. This protocol was adapted for purification in 96-well format by using AcroPrep 96 Filter Plates (Pall Life Sciences, Port Washington, NY, United States) according to the manufacturer's instructions. Lysates were clarified using Lysate Clearance plates (3 μm GxF, 0.2 μm Supor) and transferred to filter plates (0.45 μm GHP) containing 50 μl Strep-tactin resin per well. Wells were washed 3× with 50 mM Tris-HCl pH 8.5 containing 100 mM NaCl and 200 μM ZnCl$_2$ and 3× with pH 7.5 buffer. After elution, samples were concentrated and elution buffer removed using ultrafiltration plates (Omega 10K membrane).

## Kinetic characterization of variants

Paraoxon, 2NH, and Fast Red were purchased from Sigma (St. Louis, MO, United States). Substrates for linear free energy relationships were gifts from Dan Tawfik's laboratory and their synthesis is described in *Khersonsky and Tawfik (2005)*. Absorbance wavelengths and extinction coefficients are given in *Supplementary file 2*. For determination of initial rates in lysate, cells were grown and assayed in at least duplicate as described under the section 'Screens in 96-well plates'. The experiment was repeated and the average change relative to the respective parent variant and the standard deviation were determined (*Supplementary file 2*). A Student's *t*-test was performed to obtain p-values. Where applicable, p-values were also calculated to determine whether the effect of a certain mutation in two different backgrounds (rather than compared to the parent mutant lacking this mutation) is significant. For determination of initial rates using purified enzyme, variants were expressed and purified in 96-well format in at least duplicate and assayed as described above in 'Screens in 96-well plates'. For determination of Michaelis–Menten parameters, reactions were performed in triplicate at a range of substrate concentrations (0–2000 μM). Reactions were initiated by addition of 180 μl of substrate solution (in 50 mM Tris-HCl, pH 7.5 supplemented with 0.02–0.1% Triton X-100) to 20 μl of enzyme in 50 mM Tris-HCl, pH 7.5 supplemented with 200 μM ZnCl$_2$ and 0.02% Triton X-100. Data were fit to Michaelis–Menten kinetics in Kaleidagraph.

## Purification of untagged proteins for crystallization

AE and *neo*PTE genes were cloned into pET32-trx plasmid without Strep-tag using FastDigest *Nco*I and *Hind*III as described above, transformed into *E. coli* BL21 (DE3), and grown for 72 hr at 30°C in TB medium containing 100 µg/ml ampicillin and 500 µM ZnCl$_2$. Cells were harvested by centrifugation at 3320×g and 4°C for 10 min, resuspended in 20 mM Tris-HCl pH 8 containing 100 µM ZnCl$_2$ and lysed by sonication (OMNI Sonic Ruptor 400, Thermo Scientific, Waltham, MA, United States, 3× 30 s on/60 s off, amplitude 40%). Cell debris was removed by centrifugation at 30,000×g and 4°C for 45 min and lysate filtered through 45 µm filters (Millipore). The lysate was loaded onto two HiPrep Q FF columns (GE Healthcare, Wauwatosa, WI, United States) in series. PTE elutes in the flow through as well as the early wash fractions. Active fractions were pooled and passed through a 45 µm filter. The sample was concentrated over a Millipore spin column (MWCO 30,000) and purified by gel filtration (HiLoad 16/60 Superdex 200 prep grade, GE Healthcare, Wauwatosa, WI, United States). Protein was concentrated to 12 mg/ml and stored at 4°C.

## Crystallization, data collection and structure determination

Crystals of *wt*PTE, AE, and *neo*PTE were obtained by vapor diffusion from a solution containing protein (10 mg/ml) plus 20–30% wt/vol 2-methane-4-pentane diol (MPD), buffered to pH 6.5 by 0.1 M sodium cacodylate, as described previously (*Tokuriki et al., 2012*). Serial microseeding was performed to increase crystal size (*Bergfors, 2003*). Crystals grew to approximately 200 micrometers and were soaked in 40% MPD, 0.1 M sodium cacodylate as cryoprotectant for 5–10 min and then flash-cooled to 100 K in the gaseous nitrogen cryostream of a cooling device (Oxford Cryosystems, United Kingdom). Data were collected from frozen crystals on beamline MX1 of the Australian Synchrotron (AS). The data were indexed and integrated by XDS (*Kabsch, 2010*) and Aimless (*Evans and Murshudov, 2013*), with data cut-off being made at the highest resolution that retained a mean half dataset correlation coefficient (CC$^{1/2}$) of at least 0.5 in the outer shell (*Supplementary file 3*) (*Karplus and Diederichs, 2012*). Although all three crystals were crystallized in the same conditions, *neo*PTE crystallized in a different space group with different unit cell dimensions (p65, with a h, −h−k, l merohedral twin operator, vs C2221). A starting model for refinement (R18; PDB ID: 4E3T) (*Tokuriki et al., 2012*) was used to provide initial phases. Structures were refined with phenix.refine (*Afonine et al., 2012*) and validated with molprobity (*Chen et al., 2010*), as implemented in the phenix software suite (*Supplementary file 3*) (*Adams et al., 2010*).

## Acknowledgements

We thank Dan S Tawfik and members of the Tokuriki laboratory for comments on the manuscript and Kirsten Wyganowski for technical support. This work was supported by the Natural Sciences and Engineering Research Council of Canada. MK was partially supported by the EU ITN ProSA. NT is a CIHR new investigator and a Michael Smith Foundation of Health Research (MSFHR) career investigator. CJJ acknowledges an ARC Discovery Early Career Researcher Award. FH is an ERC Starting Investigator.

# Additional information

## Funding

| Funder | Grant reference | Author |
| --- | --- | --- |
| Natural Sciences and Engineering Research Council of Canada (Conseil de Recherches en Sciences Naturelles et en Génie du Canada) | Discovery Grants RGPIN 418262-12 | Nobuhiko Tokuriki |
| Australian Research Council (ARC) | FT140101059 | Colin J Jackson |
| European Commission (EC) | MRTN-CT-2005-019475 | Florian Hollfelder |

| Funder | Grant reference | Author |
|---|---|---|
| Biotechnology and Biological Sciences Research Council (BBSRC) | BB/I004327/1 | Florian Hollfelder |
| European Research Council (ERC) | 208813 | Florian Hollfelder |

The funders had no role in study design, data collection and interpretation, or the decision to submit the work for publication.

## Author contributions

MK, NT, Conception and design, Acquisition of data, Analysis and interpretation of data, Drafting or revising the article; CJJ, Acquisition of data, Analysis and interpretation of data, Drafting or revising the article; ECC, Acquisition of data, Analysis and interpretation of data; FH, Analysis and interpretation of data, Drafting or revising the article

## Additional files

### Supplementary files
• Supplementary file 1. Description of the directed evolution rounds.

• Supplementary file 2. Kinetic parameters of all variants.

• Supplementary file 3. Crystallographic information.

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
