## [Decision Letter]

Thank you for sending your work entitled “Reverse evolution leads to molecular speciation despite functional and active-site convergence” for consideration at *eLife*. Your article has been favorably evaluated by Michael Marletta (Senior Editor) and three reviewers, one of whom is a member of our Board of Reviewing Editors.

The Reviewing Editor and the reviewers discussed their comments before we reached a decision to invite you to revise your submission. As you'll see below, the reviewers were somewhat enthusiastic about the work and its potential suitability for *eLife*. However, there were a number of major concerns about the clarity and precision of the writing and discussion of your results. Although standard practice for *eLife* is to provide authors a single review that integrates the major comments from the reviewers, we have decided to provide you the reviewer comments in full. As you'll see the reviewer comments are thoughtful and detailed, so any attempt to summarize them runs the risk of obscuring or confusing their message.

Reviewer #1:

Overall, I like this manuscript and I think it presents a nice story about enzyme evolution. The data are relatively clean and straightforward. My primary concern with this manuscript is the frequent claim throughout it that evolution is “irreversible”. This isn't really accurate, I don't think. The mutations that accumulated could always be reversed in the exact, reverse order in which they initially occurred. I think what the authors really mean when they say “irreversible” is that (i) the probability of an exact reversion of multiple mutations is improbable and (ii) reversion of only some mutations may not restore the original phenotype if other mutations have also occurred. In other words, let's say evolution of *wt*PTE into AE required substitutions Z1z and Y2y (i.e. ZY at positions 1/2 = phenotype 1 and zy at positions 1/2 = phenotype 2) but that a third mutation, say X3x occurred at position 3. If that mutation is incompatible with ZY (i.e. ZYX = phenotype 1, while ZYx = dead protein) then the authors are saying evolution is irreversible. But that's not really logical. It is formally reversible, provided one also reverses the mutation at position 3. So, bottom line, I get what the authors are driving at, but their language is imprecise and, as a consequence, misleading. This issue affects much of the manuscript and needs attention.

Rephrasing the argument above, what the authors have demonstrated here is that there is redundancy, or degeneracy, in PTE catalysis, meaning that multiple genotypes can give rise to the same phenotype. Because of this redundancy, evolution is unlikely to return to the exact same genotype following selection for a new trait followed by selection for the old trait. Additionally, they have nicely documented the epistasis that occurs during their experimental evolution, showing how mutations can have very different effects on the *wt*PTE, AE, and *neo*PTE genetic backgrounds, or what they call genetic incompatibility. I think these ideas should be articulated more clearly rather than selling the idea of irreversibility, which, as noted above, doesn't seem at all logical to me.

In the subsection “Genotypic irreversibility and constraints underlying phenotypic reversion” the authors state: “Nine additional, ‘new mutations’ were needed to accomplish the phenotypic reversion.” Nine additional mutations certainly arose, but was each one of them *needed*? This statement needs to be better justified, or I missed something in a supplemental figure/table. If it's the latter, then the data should probably be better explained/laid out in the manuscript and main figures.

Reviewer #2:

The authors use directed evolution techniques to study evolutionary irreversibility and epistasis. They put an enzyme under selection pressure to evolve a new activity and then, after it has done so, reverse the selection to drive reacquisition of the ancestral activity. This allows the authors to chart the step-by-step mutational trajectories taken in both directions, to identify any genetic incompatibilities between the starting and ending states, and to use X-ray crystallography to identify causes of incompatibility.

This is a careful and detailed study. It sheds useful light on an important question in molecular evolution: the extent of epistasis during evolutionary trajectories and its potential effects on evolutionary processes, particularly evolutionary reversal to ancestral states. Numerous studies have addressed this problem computationally or at the phenotypic level, but our understanding of molecular irreversibility is limited. A previous molecular study showed that epistasis can prevent functionally important substitutions during historical evolution from being directly reversed, but that study did not show whether or how the ancestral function could be reacquired if the protein were subject to selection pressure to do so. The submitted manuscript is the first to use a directed evolution approach to address this question, and this strategy is quite appropriate for this purpose.

I believe that the paper has the potential to be a strong contribution to *eLife*. There are some significant issues in the way the argument is framed and the data presented that make the paper more confusing than it ought to be or, in a few places, imprecise or inadequately supported in its claims. It should be possible for the authors to address all these concerns. I detail these below.

Issues related to interpretation and analysis:

1) One significant limitation is that the authors have examined only a single evolutionary pathway in both directions. Decisively answering questions of contingency and determinism and repeatability and the number of paths between states requires examination of many paths, not just one. The authors' detailed characterization of the trajectory they studied does allow certain general inferences on these subjects to be drawn, but they are indirect because a single instance of evolution is studied. This does not invalidate the work, but some acknowledgement and discussion of the cautions required by this limitation are important.

2) Claims about effects of mutations are made without appropriate statistical analysis and rationale. For example, in Figure 4 and the claims based on these data, the authors use a 1.3-fold change in activity as a threshold of significance: changes smaller than this are considered neutral, while those that reduce activity by a factor larger than this (or by its reciprocal, to be precise), are considered deleterious. I don't see a justification for this threshold, which seems arbitrary, although it is very important for the conclusions. The authors must justify this choice rationally or change their claims. In this figure, the claim the authors want to make pertains to incompatibility; for such a claim to be made, there should be evidence to show that a state from PTE is deleterious in *neo*PTE, or vice versa: the evidence of a deleterious effect must be strong enough for us to rule out the possibility that the state is in fact neutral or beneficial. I see error bars, but I don't know what they represent, and I therefore don't know how confident I should be that many of these mutations are authentically deleterious (that is, that I can rule out the possibility that the true effect of the mutation is zero, but error over a small number of measurements has yielded a spurious reduction in activity.) The authors should apply an appropriate statistical procedure to keep the rate of false discoveries of incompatibility to an acceptable level, considering the large number of tests being conducted.

Similarly, in Figure 6, the authors say that R111*s* exhibits sign epistasis when tested in the two different backgrounds. It looks like one has a very small negative effect and the other a very small positive effect, and it is not obvious whether we can rule out zero (and therefore no difference in sign) for either one, based on the error bars. There are many other putative examples of sign epistasis in this figure for which it appears to be difficult to rule out different signs due to stochastic/measurement error around a near-zero effect. Again, we need a proper test or statistical characterization of confidence for all these claims. Similar problems affect many of the paper's figures.

3) The authors venture hypotheses about biophysical mechanisms underlying the observed functional effects of mutations based on X-ray crystal structures and structural models in which the various ligands have been computationally placed into the active site of the crystal structures. The conclusions about the causes of functional incompatibility are mostly based on observations of clashes observed in models between proteins and their non-preferred substrates. Proteins are flexible, however, and can often accommodate new ligands in ways that models do not predict. This possibility and the uncertainty introduced by using a model of a complex should be acknowledged.

4) The authors propose structural mechanisms for some of the functional effects of mutations that they observe: some of these involve changes in structure at the sub-angstrom level, but the structures are themselves are only at resolution of 1.6 to 2.0 angstrom. How confident can we be in such fine-scale putative differences in the location of atoms given the actual resolution of the structures? These inferences should be made with caution or not made at all.

There are some conceptual issues that I feel should be thought out more clearly and expressed more precisely:

1) The authors swap states between PTE and *neo*PTE, observe that swapping some residues severely compromises function, and conclude that the proteins occupy “separate adaptive peaks” that are not “connected” on the adaptive landscape. The thinking and language here are imprecise. All proteins are by definition connected via some number of mutations on the adaptive landscape. The question the authors seek to address whether they are accessible from each other via a continuously connected neutral (or functional) network of single-replacement changes (in the sense used by Maynard Smith and A. Wagner). Incompatibility of single mutation-swaps does not mean that proteins cannot be reached through such a connected network. If permissive mutations interact epistatically with the “incompatible” amino acid and if the permissive mutations can be introduced without deleterious effect on the function, then the genotypes with and without the “incompatible” amino acid can in fact be connected. Incompatibility does mean that the cluster of functional genotypes containing the ancestral state at the residue of interest and the functional network of genotypes containing the derived state have fewer connections than they might have if no incompatibility were present. That is all that can be said about connectivity without further analysis. The large number of incompatibilities the authors observe may indicates that the number of connections is reduced in a fairly dramatic way, but how dramatic the reduction is relative to the total number of possible connections a priori is also unknown without further analysis.

Another issue related to epistasis, sequence space and contingency is the sufficiency of selection to move a protein through sequence space from one protein to another. If the permissive mutations that make residues tolerable in one genetic background but not the other can be introduced without affecting the PTE function, then the authors' observations would show that selection for PTE function would not be sufficient to drive the reacquisition of the ancestral state at the residue of interest; in fact, this does seem to be true. There are probably other interesting and valid ways for the authors to describe their findings, but they should modify their language to make the conceptual model of an adaptive landscape and the conclusions they draw more precise.

2) According to the Abstract, selection for the ancestral function drove the acquisition of a new genotype provides evidence for irreversibility. This is not clearly thought out. If there were no epistasis and no selective barriers to reacquisition of the ancestral genotype (or any other genotype that confers the ancestral function), and so long as there were numerous genotypes that encode the ancestral function, then selection for the ancestral phenotype is likely to produce different outcomes every time. That is, the simple fact that under selection the genotype that evolved is different from the ancestral genotype does not establish epistasis, incompatibility, or irreversibility. It is the authors' other experiments that show this.

3) The authors say that selection for aryl esterase activity was accompanied by a decrease on PTE activity, which indicates an “intrinsic trade-off.” This conclusion is not correct. If most random mutations reduce PTE activity and there is no necessary or intrinsic mechanistic trade-off between PTE and AE activity, then mutations that increase AE activity are more likely than not to reduce PTE activity.

4) The authors use a metaphor of speciation for the evolution of incompatible enzyme genotypes. In the Discussion, however, it appears that they use the term literally, as if the evolution of this enzyme could cause reproductive isolation between populations of organisms. There is no evidence to support this causal leap, and the authors should avoid this type of discussion of the issue. If they are making an analogy between reproductive isolation of populations due to epistasis/Dobzhansky-Muller effect and the evolution of incompatible amino acids in their PTE enzymes that is fine, but the analogy should be labeled as such – as a metaphor. The authors say that evolution of the genotype is “constrained” to follow certain paths deterministically under selection for the ancestral function, and they say that Harms et al. “showed that the evolution of hormone receptors is similarly constrained by functional requirements imposed on the binding pocket.” The comparison seems imprecise. The Harms paper argued that evolution of the protein does not proceed deterministically, because function-changing mutations require prior rare mutations that do not change the function; the rarity of these permissive mutations comes from the fact they must fulfill numerous requirements. The authors are comparing this case to their own, in which they say the evolution of certain aspects of the ancestral genotype is deterministic because they represent the only way to achieve the ancestral function. The language should be cleaned up here.

5) The authors use the word constraint in a way that I find confusing. (This is not surprising, because the term is used variously and inconsistently in the literature, too.) Here, the authors say that catalytic requirements impose evolutionary constraints that virtually guarantee that something like the ancestral structure will re-evolve when selection for the ancestral function is imposed. This usage seems inconsistent with the way the term constraint has usually been used. Although the term has numerous meanings, it is almost always used to describe requirements that slow the rate of evolution or limit the capacity of selection to produce an outcome: 1) “functional constraints,” which refers to limitations on the process of genetic drift that are imposed by purifying selection and thus limit the rate of evolution; 2) “developmental (and similar) constraints,” which refers to limits on the kinds of variation that mutation can produce and make available selection; or 3) biochemical constraints or trade-offs, which make it nearly impossible for mutation and selection to produce an outcome that is “optimal” for some property looked at in isolation. The authors' point seems to be that there is one easily accessible way of solving the “problem” of esterase activity imposed by the selection pressure, so evolution is likely to always produce this outcome. This doesn't seem to me to be a constraint but rather a lack of constraint on the capacity of mutation and selection to produce exactly what the experimenters have asked it to do. I suggest finding another word.

Reviewer #3:

This manuscript describes the evolution of a new arylesterase activity in a phosphotriesterase, and then a reversal to the phenotype of the ancestral enzyme. The central question is whether the trajectory toward the new activity is reversible.

The authors report beautiful structural data showing how the active site of the enzyme can be reshaped to allow an efficient new activity, and how it can be reshaped again to restore the initial activity. The manuscript includes a vast amount of kinetic data for enzymes at various stages of the forward and reverse trajectories. The work is fascinating and of high quality. It is particularly intriguing that the final enzyme had very high phosphotriesterase and arylesterase activity.

Major comments:

1) I feel that the Introduction is somewhat overblown. We know from the structures of orthologous enzymes that there are many ways (in terms of primary sequence) to achieve the end result of an efficient enzyme when starting from a common ancestor. We also know that epistasis is important. Thus, the demonstration that the reverse evolution toward the original activity proceeds via a different trajectory does not seem surprising to me. In fact, I would be surprised if the evolutionary trajectory had been perfectly reversible.

2) I object to the use of terms such as speciation and genetic incompatibility that apply to reproductive isolation of organisms to describe enzyme variants. Enzymes do not reproduce or exchange residues. These terms should be replaced with something more appropriate for molecular evolutionary processes.

3) The authors should discuss the fact that they have explored only one possible trajectory for both the forward and reverse evolution stages, and that there are likely to be others. In fact, there may be trajectories that are fully reversible in terms of genotype. A fair statement from this one example is that reversion of a phenotype does not require reversion of a genotype.

4) In the subsection “The active site converged to its original state in the reverse evolution”, the authors say that “the rate-limiting step of phosphotriester hydrolysis was changed in AE, but restored in *neo*PTE” (Figure 2—figure supplement 3)”. This point requires an expanded discussion in the text, which should explain how the experiment was done, why the results shown in the supplementary material indicate that the rate-limiting step has changed, and what the rate-limiting step might be in each case.

5) Beginning with the discussion of Figure 5, the manuscript is very difficult to follow due to imprecise wording and obscure terminology.

A) The presentation of the data in Figure 5 is very hard to follow: “Underlined values indicate that the amino acid in question (Thr in this case) is already present in a certain background, and the effect is calculated as its reintroduction after removal (reversion of *wt*PTE-t45A to *wt*PTE).” So some effects are measured in one direction and some in the other, I think. The authors should find a better way to communicate these data. They did an enormous amount of work and the results are important.

B) I could not figure out the second paragraph of the subsection “Mutational epistasis underlies genotypic irreversibility” even after spending considerable time trying. The results need to be described in more precise language in order to avoid confusing the reader. For example, the passage: “the effect of mutations is significantly altered by the accumulation of subsequent mutations”. How can the effect of a mutation be altered by something that has not yet occurred?

C) Please clarify: “reversions that were initially deleterious for phosphotriesterase activity became favorable in the reverse evolution”. During the forward evolution, mutations occurred (not reversions), and the mutations were deleterious for phosphotriesterase activity.

D) The authors state: “all new mutations had no effect or a negative effect on phosphotriesterase activity at the onset of forward evolution”. Figure 1 shows that the first mutation that occurred decreased phosphotriesterase activity.

E) Figure 7 is also difficult to understand. The mutations are listed as, for example, I341*t*. To me, that suggests reversion of I341 to the ancestral Thr. But looking at Figure 1 tells me that the ancestral residue was Ile, and that the mutant enzymes (AE and *neo*PTE) have Thr at that position. Figure 7 also shows the effect of V144*e*, which is even more confusing because Figure 1 indicates that the ancestral residue was Thr and it changed to Val.

F) In the subsection “Mutational epistasis underlies genotypic irreversibility”, the sentence “The replacement of *f*306I by *s*308c…” does not make sense.

[Editors' note: further revisions were requested prior to acceptance, as described below.]

Thank you for resubmitting your work entitled “Reverse evolution leads to molecular speciation despite functional and active-site convergence” for further consideration at *eLife*. Your revised article has been favorably evaluated by Michael Marletta (Senior Editor), a Reviewing Editor, and two reviewers. The manuscript has been improved but there are some remaining issues that need to be addressed. These issues are listed below. Each item needs to be fully addressed if a re-revised manuscript is to be considered further for publication.

Reviewer #2:

1) I remain concerned about the threshold of 1.3-fold change. An increase in activity of 1.3-fold by a mutation “is considered significant.” The authors say that clones displaying a 1.3-fold improvement are generally verified as authentically displaying an increase in activity upon repeat examination. There are several issues about which I am concerned here. The first is that no data are provided to support this justification for the threshold; in fact, it is never articulated in the paper, although it is found in the letter. Thus the 1.3-fold threshold remains unjustified in the text. Second, the use of this threshold in defining examples of epistasis strikes me as problematic. Mutations that produce an observed increase in activity by >1.3x in one background but not another are judged to be epistatically modified. Thus, an increase of 1.4x in one background but 1.2x in another exhibit epistasis, because they are significantly improved in the first case but not the second. The authors do not present any way to establish whether the differences between these two genotypes are really statistically or biologically significant, so the claim of epistasis remains unjustified. The authors argue that a statistical analysis shows that most cases of 1.3x improvement are statistically significant (also not shown), but this is beside the point; the relevant issue is whether the *difference* between the effects of a mutation in two genetic backgrounds is statistically and biologically significant, not whether a 1.3-fold effect one mutation is different from no effect. Many, although not all, of the authors' examples of epistasis in the paper are, in fact, rather subtle examples of phenomena like this – not sign epistasis, or very large effects vs. very small or no effects, but the difference between an effect slightly greater than 1.3 and one slightly less. I therefore believe that the authors need to strengthen this aspect of the analysis. I recommend a test of differences of effect between backgrounds.

2) The authors make inferences about clashes on a sub-angstrom scale from structures that are only at 1.6 to 2 angstrom resolution. They say that the B-factors are low, but this is of limited relevance. This indicates that there is limited thermal motion for these atoms, but the resolution of the structure still does not allow location of the atom with precision greater than the stated resolution, which is necessary to strongly support the claims made. I believe this could be dealt with in the manuscript with more direct cautionary language and softening of the claims.

3) The idea of different protein “species” that are “genetically incompatible” with each other is still obscure, and I do not find it persuasive. Do the authors mean that if the alleles were brought together in a heterozygote that recombination within the gene would produce nonfunctional proteins and might be selected against? Are they arguing that this might contribute to speciation? If that's what the authors mean, they should say so, but it seems very far-fetched and marginally relevant to our understanding of protein evolution. I think the effects of epistasis on the evolutionary potential of a protein are a much more solid basis for the authors' interpretation.

Reviewer #3:

1) The “speciation” concept that another reviewer and I both objected to is still present in the title, Abstract and text. I think alternative wording should be used whenever possible, and the title should be changed.

2) The Introduction is still somewhat grandiose. In particular, I am bothered by the first sentence. I do not see how reversibility relates to the question of whether re-playing life's tape would lead to the same outcome. The latter question addresses whether the same things would happen again starting from the same place, while the former asks whether an individual trajectory can be reversed. Either omit or rephrase/expand to clarify the connection.

3) Is there any reason not to plot *k*_cat_/*K*_M_ values in Figure 1? They would be more relevant in terms of the changes in the two enzyme activities as the evolution proceeds.

4) The section head in the Results section states that “The active site converged to its original state in the reverse evolution”. I feel that this is misleading. The active site residues were not restored, and the shape of the active site is rather different. It would be correct to say that a functional active site had been restored, but I don't agree with the claim in this section (and elsewhere in the manuscript) that the active site converged to its original state. Further, the authors say that “the naphthyl binding pocket remains intact […] which likely explains why *neo*PTE is still bi-functional”. This statement is not consistent with the statement that the active site converged to its original state.

5) In the aforementioned subsection, it is claimed that: “the number of accessible mutational trajectories that lead to a wild-type level fitness peak from AE are highly limited”. I don't believe this statement is justified. There may well be many trajectories that could lead to *wt*-level activity that start from states that were discarded during the evolution because they were not the best variants at that particular stage.

6) A picture showing smooth and rugged fitness landscapes would be useful for readers who are not experts in molecular evolution.

7) Subsection “Emergence of incompatibility between the two seemingly identical enzymes”: the heading of this section refers to “two seemingly identical enzymes”. The enzymes are not seemingly identical, as *neo*PTE still has fairly robust arylesterase activity.

8) The terminology used to describe mutations in the discussion of Figures 5 and 6 is problematic. Usually the authors use the conventional notation; e.g. AxxB, which means that A at position xx was changed to B. But sometimes they use the reverse. For example, in the subsection “Mutational epistasis underlies genotypic irreversibility”, the text says: “some mutations initially increased (I172*t* and F271*l*) or were neutral (V130*l*) to phosphotriesterase activity…”. The mutations during the forward evolution were actually *t*172I, *l*271F and *l*130V. According to Figure 1 and Figure 1—figure supplement 1, *t*172I decreased, rather than increased, phosphotriesterase activity. This section needs to be clarified.

9) Figure 5 is still nearly incomprehensible to me. For example, in the column labeled “R254*h*”, the legend says that the effect is calculated in the direction R254*h* in all cases. So the figure suggests that changing R254 to H increases phosphotriesterase activity by 13-fold. But the *wt* enzyme has *His* at 254, not *Arg*. So what is really meant is that changing *His*254 to *Arg* decreases phosphotriesterase activity by 13-fold, which is consistent with Figure 1. Having to do these mental gymnastics for each position and in each background makes my brain hurt. I do not think this way of conveying the data is salvageable.

10) Figure 6 shows the effects of V130*l* on phosphotriesterase activity in the “forward evolution” and “AE background” contexts. Figure 6 would be easier to understand if it specified V130*l* in R13*b*, and *I*172T in R5*a* (and so on), rather than using the term “forward evolution”, which initially gave me the impression that V130 was changed to I during the forward evolution. In fact, I am still confused by whether the bars at the left side of Figure 6 refer to the actual mutation that occurs during the forward evolution, or the reversion of the mutation at position 130. Even after several tries, I am unable to follow the discussion in subsection “Mutational epistasis underlies genotypic irreversibility”.

[Editors' note: further revisions were requested prior to acceptance, as described below.]

Thank you for resubmitting your work entitled “Reverse evolution leads to genotypic incompatibility despite functional and active-site convergence” for further consideration at *eLife*. Your revised article has been favorably evaluated by Michael Marletta (Senior Editor) and a Reviewing Editor. The manuscript has been improved but there is one remaining issue that needs to be addressed before acceptance:

An issue raised by several of the reviewers during both rounds of revision related to the terms ‘molecular speciation’ and ‘incompatibility’. While you have removed the confusing former term, the latter still appears in several places in the text. This is fine, but it needs, in some instances, a bit of clarification and adjustment of the language. For instance, the Abstract states that the reverse evolution experiment performed led “to a different sequence incompatible with the original one, despite functional identity.” As written, the statement is ambiguous and confusing in the sense that what you really mean is that certain substitutions that occurred during the reverse evolution process are incompatible with some of the residues present in the ancestral/original protein. It's not that the whole sequence is somehow incompatible with the original one, as written. Please adjust the language throughout the text where this concept arises to improve precision and meaning.

---

## [Author Response]

Reviewer #1:

*Overall, I like this manuscript and I think it presents a nice story about enzyme evolution. The data are relatively clean and straightforward. My primary concern with this manuscript is the frequent claim throughout it that evolution is “irreversible”. This isn't really accurate, I don't think. The mutations that accumulated could always be reversed in the exact, reverse order in which they initially occurred. I think what the authors really mean when they say “irreversible” is that (i) the probability of an exact reversion of multiple mutations is improbable and (ii) reversion of only some mutations may not restore the original phenotype if other mutations have also occurred. In other words, let's say evolution of* wt*PTE into AE required substitutions Z1z and Y2y (i.e. ZY at positions 1/2 = phenotype 1 and zy at positions 1/2 = phenotype 2) but that a third mutation, say X3x occurred at position 3. If that mutation is incompatible with ZY (i.e. ZYX = phenotype 1, while ZYx = dead protein) then the authors are saying evolution is irreversible. But that's not really logical. It is formally reversible, provided one also reverses the mutation at position 3. So, bottom line, I get what the authors are driving at, but their language is imprecise and, as a consequence, misleading. This issue affects much of the manuscript and needs attention*.

What we mean when we say “evolution is genotypically irreversible” is that a certain mutation cannot be reverted when a selection pressure to restore the original function is applied. First, if all mutations accumulated in the forward trajectory would be either detrimental or neutral to phosphotriesterase activity, it would certainly be the case that exact reversal of the mutations from rounds 22 back to 1 would lead to a theoretically possible trajectory. However, some of the forward mutations actually increased phosphotriesterase activity, meaning their reversal would not be possible. Second, as soon as new mutations occur, they negate the effect of some reversions, making even these initially possible changes impossible.

In the example given by the reviewer, mutation X3x could be such a new mutation occurring in the reverse trajectory: because it increases phosphotriesterase activity, it cannot be reverted. If it interacts with the amino acids in positions 1 and 2 and negates the effect of reversions z1Z and y2Y, these transitions will be blocked. Alternatively, X3x could also be a mutation that occurred in the forward trajectory and increased phosphotriesterase activity, in which case it could also not be reverted. Again, if it shows sign-epistasis with the other two positions, they cannot be reverted either. Therefore, we argue that irreversibility and incompatibility are strongly connected. In our original manuscript, we describe these effects (subsections “Phenotypic reversibility in the laboratory evolution of PTE”, “Genotypic irreversibility and constraints underlying phenotypic reversion” and “Mutational epistasis underlies genotypic irreversibility”). However, we agree with the reviewer that this should be emphasized and better explained, and have therefore added a summary to the Discussion (“Genotypic irreversibility was caused by several factors […] the trajectory deviates further from the original path.”).

*Rephrasing the argument above, what the authors have demonstrated here is that there is redundancy, or degeneracy, in PTE catalysis, meaning that multiple genotypes can give rise to the same phenotype. Because of this redundancy, evolution is unlikely to return to the exact same genotype following selection for a new trait followed by selection for the old trait. Additionally, they have nicely documented the epistasis that occurs during their experimental evolution, showing how mutations can have very different effects on the* wt*PTE, AE, and* neo*PTE genetic backgrounds, or what they call genetic incompatibility. I think these ideas should be articulated more clearly rather than selling the idea of irreversibility, which, as noted above, doesn't seem at all logical to me*.

We appreciate this suggestion and have rewritten the Abstract and Introduction accordingly. The revised version puts more emphasis on genotypic incompatibility while the idea of irreversibility has been toned down. It also includes the term “genotypic redundancy” throughout the manuscript (Introduction, Results and Discussion).

*In the subsection “Genotypic irreversibility and constraints underlying phenotypic reversion” the authors state: “Nine additional ‘new mutations’ were needed to accomplish the phenotypic reversion.” Nine additional mutations certainly arose, but was each of one of them* needed*? This statement needs to be better justified, or I missed something in a supplemental figure/table. If it's the latter, then the data should probably be better explained/laid out in the manuscript and main figures*.

We have rephrased the sentence (“Nine additional, ‘new mutations’ accumulated…”).

Reviewer #2:

[…] Issues related to interpretation and analysis:

1) One significant limitation is that the authors have examined only a single evolutionary pathway in both directions. Decisively answering questions of contingency and determinism and repeatability and the number of paths between states requires examination of many paths, not just one. The authors' detailed characterization of the trajectory they studied does allow certain general inferences on these subjects to be drawn, but they are indirect because a single instance of evolution is studied. This does not invalidate the work, but some acknowledgement and discussion of the cautions required by this limitation are important.

We agree with the reviewers (Reviewer 3 has also raised this point). First, we would like to note that we explored a second trajectory where we prevented the fixation of reversions (see last paragraph of subsection “The active site converged to its original state in the reverse evolution” and the second paragraph of the Discussion). This trajectory (1) plateaued at a phosphotriesterase activity level far below that of *wt*PTE and (2) consisted largely of the same new mutations that also accumulated in the main trajectory. This gives strong support for the observations made in the main trajectory that (1) some reversions are necessary for phenotypic reversibility, i.e. appear repeatedly and (2) certain new mutations also appear repeatedly, which shows that evolution is constrained by the necessity to realize a particular active-site configuration. Nevertheless, we fully agree that multiple parallel evolutionary experiments would be the method of choice to gain more general insights and strengthen these claims. Therefore, we have added the following to the Discussion:

“Another important limitation of our work is that we only examined two evolutionary trajectories (the main trajectory and the trajectory without reversions). One could imagine conducting multiple parallel evolutionary experiments to shed light on the repeatability of the trajectory taken, but our screening system is not amenable to such a throughput.”

*2) Claims about effects of mutations are made without appropriate statistical analysis and rationale. For example, in*
Figure 4
*and the claims based on these data, the authors use a 1.3-fold change in activity as a threshold of significance: changes smaller than this are considered neutral, while those that reduce activity by a factor larger than this (or by its reciprocal, to be precise), are considered deleterious. I don't see a justification for this threshold, which seems arbitrary, although it is very important for the conclusions. The authors must justify this choice rationally or change their claims. In this figure, the claim the authors want to make pertains to incompatibility; for such a claim to be made, there should be evidence to show that a state from PTE is deleterious in* neo*PTE, or vice versa: the evidence of a deleterious effect must be strong enough for us to rule out the possibility that the state is in fact neutral or beneficial. I see error bars, but I don't know what they represent, and I therefore don't know how confident I should be that many of these mutations are authentically deleterious (that is, that I can rule out the possibility that the true effect of the mutation is zero, but error over a small number of measurements has yielded a spurious reduction in activity.) The authors should apply an appropriate statistical procedure to keep the rate of false discoveries of incompatibility to an acceptable level, considering the large number of tests being conducted*.

*Similarly, in*
Figure 6*, the authors say that R111*s *exhibits sign epistasis when tested in the two different backgrounds. It looks like one has a very small negative effect and the other a very small positive effect, and it is not obvious whether we can rule out zero (and therefore no difference in sign) for either one, based on the error bars. There are many other putative examples of sign epistasis in this figure for which it appears to be difficult to rule out different signs due to stochastic/measurement error around a near-zero effect. Again, we need a proper test or statistical characterization of confidence for all these claims. Similar problems affect many of the paper's figures*.

We thank the reviewer for this comment and apologize for the lack of explanation provided on this point. We chose a 1.3-fold improvement as cut-off throughout the directed evolution experiment because in our experience, clones with initial improvements of >1.3-fold were consistently confirmed as improved variants, whereas smaller improvements often turned out to be false positives. Because this threshold was applied as selection threshold, we used it to evaluate all other mutants generated as well.

However, as the reviewer suggested, we have re-analyzed the lysate data to determine *p*-values. The data was calculated from two biological replicates (independent bacterial growth, protein expression, etc.), each containing 2-3 technical replicates (4-6 samples in total). We find that a 1.3-fold change is consistently significant (in some cases, smaller changes are also significant). Therefore, we can rule out the reviewer’s concern that changes >1.3-fold are neutral instead of authentically deleterious.

*3) The authors venture hypotheses about biophysical mechanisms underlying the observed functional effects of mutations based on X-ray crystal structures and structural models in which the various ligands have been computationally placed into the active site of the crystal structures. The conclusions about the causes of functional incompatibility are mostly based on observations of clashes observed in models between proteins and their non-preferred substrates. Proteins are flexible, however, and can often accommodate new ligands in ways that models do not predict. This possibility and the uncertainty introduced by using a model of a complex should be acknowledged*.

We thank the reviewer for raising this important point. The active sites of enzymes, such as PTE, are indeed flexible. However, the observation of some level of substrate turnover suggests that the substrate can bind and undergo hydrolysis at least some of the time. Because of this, analysis of the dominant structure observed in the crystal can be informative, since it gives an indication of the state most often adopted by the enzyme. Likewise, modeling the ligand into the active site, based on solved crystal structures, needs to be interpreted with some caution, as the existence of alternative binding modes cannot be excluded. Nevertheless, it strongly suggests that a number of the mutations that have accumulated increase steric hindrance for correct binding of paraoxon in AE, and in reverse for the *neo*PTE variant.

To acknowledge this level of uncertainty, we have added a list of caveats to the subsection “The active site converged to its original state in the reverse evolution”.

*4) The authors propose structural mechanisms for some of the functional effects of mutations that they observe: some of these involve changes in structure at the sub-angstrom level, but the structures are themselves are only at resolution of 1.6 to 2.0 angstrom. How confident can we be in such fine-scale putative differences in the location of atoms given the actual resolution of the structures? These inferences should be made with caution or not made at all*.

The observed changes are indeed subtle, at the sub-angstrom scale. However, the active site of these proteins superimpose almost perfectly and the atoms of the active site have some of the lowest B-factors in the protein, suggesting that the positions of the atoms can be predicted with some certainty (we have added this explanation to the subsection “The active site converged to its original state in the reverse evolution”). We have added a supplement to Figure 2 (Figure 2—figure supplement 2) showing an overlay of the electron density maps of *wt*PTE, AE, and *neo*PTE to support this claim.

*There are some conceptual issues that I feel should be thought out more clearly and expressed more precisely*:

*1) The authors swap states between PTE and* neo*PTE, observe that swapping some residues severely compromises function, and conclude that the proteins occupy “separate adaptive peaks” that are not “connected” on the adaptive landscape. The thinking and language here are imprecise. All proteins are by definition connected via some number of mutations on the adaptive landscape. The question the authors seek to address whether they are accessible from each other via a continuously connected neutral (or functional) network of single-replacement changes (in the sense used by Maynard Smith and A. Wagner). Incompatibility of single mutation-swaps does not mean that proteins cannot be reached through such a connected network. If permissive mutations interact epistatically with the “incompatible” amino acid and if the permissive mutations can be introduced without deleterious effect on the function, then the genotypes with and without the “incompatible” amino acid can in fact be connected. Incompatibility does mean that the cluster of functional genotypes containing the ancestral state at the residue of interest and the functional network of genotypes containing the derived state have fewer connections than they might have if no incompatibility were present. That is all that can be said about connectivity without further analysis. The large number of incompatibilities the authors observe may indicates that the number of connections is reduced in a fairly dramatic way, but how dramatic the reduction is relative to the total number of possible connections a priori is also unknown without further analysis*.

We agree with the reviewer that a more careful and precise description of the connectivity of *wt*PTE and *neo*PTE on the adaptive landscape is necessary. We had initially described their connectivity as follows: “It remains unknown whether the two species comprise separate peaks on the fitness landscape or are connected by mutational ridges.” To clarify, we have now added the following: “It remains unknown whether the two species comprise separate peaks on the fitness landscape or are connected through a neutral network, i.e. if the neutral exchanges would permit the subsequent occurrence of initially deleterious exchanges.”

In addition, we have rephrased all other instances where adaptive peaks were mentioned.

*Another issue related to epistasis, sequence space and contingency is the sufficiency of selection to move a protein through sequence space from one protein to another. If the permissive mutations that make residues tolerable in one genetic background but not the other can be introduced without affecting the PTE function, then the authors' observations would show that selection for PTE function would not be sufficient to drive the reacquisition of the ancestral state at the residue of interest; in fact, this does seem to be true. There are probably other interesting and valid ways for the authors to describe their findings, but they should modify their language to make the conceptual model of an adaptive landscape and the conclusions they draw more precise*.

As mentioned in our reply to the reviewer’s previous comment, we have modified the way we describe the connectivity of *wt-* and *neo*PTE on the adaptive landscape. Regarding the possibility of neutral mutations occurring first that epistatically interact with other, initially deleterious mutations, making them neutral and therefore tolerable, we are now describing this using the concept of a “neutral network” (subsection “Emergence of incompatibility between the two seemingly identical enzymes”).

*2) According to the Abstract, selection for the ancestral function drove the acquisition of a new genotype provides evidence for irreversibility. This is not clearly thought out. If there were no epistasis and no selective barriers to reacquisition of the ancestral genotype (or any other genotype that confers the ancestral function), and so long as there were numerous genotypes that encode the ancestral function, then selection for the ancestral phenotype is likely to produce different outcomes every time. That is, the simple fact that under selection the genotype that evolved is different from the ancestral genotype does not establish epistasis, incompatibility, or irreversibility. It is the authors' other experiments that show this*.

We thank the reviewer for pointing this out and have rewritten the Abstract accordingly.

*3) The authors say that selection for aryl esterase activity was accompanied by a decrease on PTE activity, which indicates an “intrinsic trade-off.” This conclusion is not correct. If most random mutations reduce PTE activity and there is no necessary or intrinsic mechanistic trade-off between PTE and AE activity, then mutations that increase AE activity are more likely than not to reduce PTE activity*.

We point out in the discussion of the crystal structures (subsection “The active site converged to its original state in the reverse evolution”):

“The structural comparison indicates that AE adapted to the planar substrate 2NH in the forward evolution, but that this came at a cost of phosphotriesterase activity, as the bulky paraoxon is no longer efficiently recognized (Figure 2).”

We then give a detailed explanation of the factors leading to this intrinsic, mechanistic trade-off (in subsections “The active site converged to its original state in the reverse evolution” and “Mutational epistasis underlies genotypic irreversibility”, Figure 2, and Figure 2—figure supplement 1): Steric hindrance, loss of shape complementarity, changes in hydrophobicity, and π-π-stacking. Therefore, it seems that the reduction of phosphotriesterase activity is a side product of the increase in arylesterase activity.

However, we agree with the referee that this does not prove that there are “intrinsic” trade-offs between the two activities, and there is the possibility that the enzyme active site can adapt to both activities simultaneously. Therefore we removed this wording from the section in question (“Phenotypic reversibility in the laboratory evolution of PTE”).

*4) The authors use a metaphor of speciation for the evolution of incompatible enzyme genotypes. In the Discussion, however, it appears that they use the term literally, as if the evolution of this enzyme could cause reproductive isolation between populations of organisms. There is no evidence to support this causal leap, and the authors should avoid this type of discussion of the issue. If they are making an analogy between reproductive isolation of populations due to epistasis/Dobzhansky-Muller effect and the evolution of incompatible amino acids in their PTE enzymes that is fine, but the analogy should be labeled as such – as a metaphor*.

Reviewer 3 has also raised this point. We agree with both reviewers that we went overboard in our usage of terminology from speciation and have cleaned up our language accordingly.

*The authors say that evolution of the genotype is “constrained” to follow certain paths deterministically under selection for the ancestral function, and they say that Harms et al. “showed that the evolution of hormone receptors is similarly constrained by functional requirements imposed on the binding pocket.” The comparison seems imprecise. The Harms paper argued that evolution of the protein does not proceed deterministically, because function-changing mutations require prior rare mutations that do not change the function; the rarity of these permissive mutations comes from the fact they must fulfill numerous requirements. The authors are comparing this case to their own, in which they say the evolution of certain aspects of the ancestral genotype is deterministic because they represent the only way to achieve the ancestral function. The language should be cleaned up here*.

The Harms paper showed that biophysical requirements dictate the limited availability of permissive mutations. In their case, evolution becomes contingent under an adaptive selection pressure because the permissive mutations are neutral by themselves, but without fixation of these permissive mutations, functional adaptation cannot occur. In the Discussion, we argue that the similarity between our and Harms’ work is the strong genetic constraint on accessible mutations dictated by biophysical requirements. We did not argue that biophysical requirements result in either “contingency” or “determinism” under an adaptive selection pressure. To clarify this, we have rephrased the sentence as follows (Discussion, second paragraph):

“Recent work by Harms et al. showed that the accessibility of functional and permissive mutations on hormone receptors is also strongly constrained by biophysical requirements imposed on the binding pocket as well as by protein dynamics (Harms and Thornton).”

*5) The authors use the word constraint in a way that I find confusing. (This is not surprising, because the term is used variously and inconsistently in the literature, too.) Here, the authors say that catalytic requirements impose evolutionary constraints that virtually guarantee that something like the ancestral structure will re-evolve when selection for the ancestral function is imposed. This usage seems inconsistent with the way the term constraint has usually been used. Although the term has numerous meanings, it is almost always used to describe requirements that slow the rate of evolution or limit the capacity of selection to produce an outcome: 1) “functional constraints,” which refers to limitations on the process of genetic drift that are imposed by purifying selection and thus limit the rate of evolution; 2) “developmental (and similar) constraints,” which refers to limits on the kinds of variation that mutation can produce and make available selection; or 3) biochemical constraints or trade-offs, which make it nearly impossible for mutation and selection to produce an outcome that is “optimal” for some property looked at in isolation. The authors' point seems to be that there is one easily accessible way of solving the “problem” of esterase activity imposed by the selection pressure, so evolution is likely to always produce this outcome. This doesn't seem to me to be a constraint but rather a lack of constraint on the capacity of mutation and selection to produce exactly what the experimenters have asked it to do. I suggest finding another word*.

We agree that our use of the term “constraint” had multiple definitions and was therefore confusing. In the revised manuscript, we have restricted it to “genotypic constraint”, or the “restrictions on accessible functional mutations”. This definition is different from what the reviewer describes as “evolutionary constraint”, but we acknowledge that many previous papers use “genetic constraint” similarly to our definition (e.g. (Weinreich et al., 2005), (7), (Taute et al., 2014), and (Podgornaia and Laub, 2015)). We have changed the term accordingly. The term “biophysical constraints” has been changed to “biophysical requirements” throughout the manuscript. Moreover, the sentence “To minimize constraints on PTE evolution caused by limited protein stability, we used GroEL/ES overexpression to buffer the destabilizing effect of mutations as previously reported […]” has been changed to: “To buffer the destabilizing effects of functional mutations and miminize reductions in soluble protein expression levels, we used GroEL/ES overexpression as previously described […]”.

Reviewer #3:

*1) I feel that the Introduction is somewhat overblown. We know from the structures of orthologous enzymes that there are many ways (in terms of primary sequence) to achieve the end result of an efficient enzyme when starting from a common ancestor. We also know that epistasis is important. Thus, the demonstration that the reverse evolution toward the original activity proceeds via a different trajectory does not seem surprising to me. In fact, I would be surprised if the evolutionary trajectory had been perfectly reversible*.

We have rewritten the Introduction (in particular the first paragraph) and toned down the description of genotypic irreversibility and contingency vs. determinism.

*2) I object to the use of terms such as speciation and genetic incompatibility that apply to reproductive isolation of organisms to describe enzyme variants. Enzymes do not reproduce or exchange residues. These terms should be replaced with something more appropriate for molecular evolutionary processes*.

Reviewer 2 has also raised this point. We agree with both reviewers that we went overboard in our usage of terminology from speciation and have cleaned up our language accordingly (please see our joint reply to Reviewer 2).

However, the term “incompatibility” has been expanded to include the protein level by the community (e.g. “protein sequence incompatibility” (56), “incompatible mutations” (37), “Dobzhansky-Mueller incompatibility” (33)). Therefore, we believe that “genotypic incompatibility”, as we call it, adequately describes the relationship between the different PTE sequences.

*3) The authors should discuss the fact that they have explored only one possible trajectory for both the forward and reverse evolution stages, and that there are likely to be others. In fact, there may be trajectories that are fully reversible in terms of genotype. A fair statement from this one example is that reversion of a phenotype does not require reversion of a genotype*.

Reviewer 2 has also raised this point. Please see our joint reply.

*4) In the subsection “The active site converged to its original state in the reverse evolution”, the authors say that “the rate-limiting step of phosphotriester hydrolysis was changed in AE, but restored in* neo*PTE” (*Figure 2—figure supplement 3*)”. This point requires an expanded discussion in the text, which should explain how the experiment was done, why the results shown in the supplementary material indicate that the rate-limiting step has changed, and what the rate-limiting step might be in each case.*

We have added more explanation to describe this experiment (“Furthermore, we measured linear free energy relationships for *wt*PTE, AE, and *neo*PTE […]. In *neo*PTE, the pattern characteristic for *wt*PTE was restored.”

However, while we can state that the rate-limiting step did not change in AE as described above, we cannot conclude from our data what the rate-limiting step may be. The same is true for the other cases. Therefore, we prefer to limit the interpretation of this experiment to comparing the overall pattern of the linear free energy relationships of the different variants.

*5) Beginning with the discussion of*
Figure 5*, the manuscript is very difficult to follow due to imprecise wording and obscure terminology*.

*A) The presentation of the data in*
Figure 5
*is very hard to follow: “Underlined values indicate that the amino acid in question (Thr in this case) is already present in a certain background, and the effect is calculated as its reintroduction after removal (reversion of* wt*PTE-t45A to* wt*PTE).” So some effects are measured in one direction and some in the other, I think. The authors should find a better way to communicate these data. They did an enormous amount of work and the results are important.*

We have removed this sentence and replaced it by a stepwise explanation of one example mutation, hoping this will make the data more clear. Because Reviewer 2 also raised this point, please see our reply above.

B) I could not figure out the second paragraph of the subsection “Mutational epistasis underlies genotypic irreversibility” even after spending considerable time trying. The results need to be described in more precise language in order to avoid confusing the reader. For example, the passage: “the effect of mutations is significantly altered by the accumulation of subsequent mutations”. How can the effect of a mutation be altered by something that has not yet occurred?

We apologize for this confusion. We have attempted to clean this up and rephrase difficult sentences to address points B) to F) of the reviewer’s comments. This now reads: “In the forward evolution, the effect of mutations is significantly altered after their fixation due to epistasis caused by mutations subsequently accumulated in the trajectory.”

*C) Please clarify: “reversions that were initially deleterious for phosphotriesterase activity became favorable in the reverse evolution”. During the forward evolution, mutations occurred (not reversions), and the mutations were deleterious for phosphotriesterase activity*.

This has been clarified in the subsection “Mutational epistasis underlies genotypic irreversibility” (“For example, some mutations initially increased […] in the background of AE (Figure 6)”).

D) The authors state: “all new mutations had no effect or a negative effect on phosphotriesterase activity at the onset of forward evolution”. Figure 1 shows that the first mutation that occurred decreased phosphotriesterase activity.

We have rephrased this sentence.

*E)*
Figure 7
*is also difficult to understand. The mutations are listed as, for example, I341*t*. To me, that suggests reversion of I341 to the ancestral Thr. But looking at*
Figure 1
*tells me that the ancestral residue was Ile, and that the mutant enzymes (AE and* neo*PTE) have Thr at that position.*
Figure 7
*also shows the effect of V144*e*, which is even more confusing because*
Figure 1
*indicates that the ancestral residue was Thr and it changed to Val*.

We have corrected these mistakes (T341*i* in Figure 7 and *e*144V in Figure 1).

*F) In the third paragraph of subsection “Mutational epistasis underlies genotypic irreversibility”, the sentence “The replacement of* f*306I by* s*308C…” does not make sense*.

We have changed this to say: “The redundancy of the mutations *f*306I and *s*308C was also evidenced by combinatorial mutational analysis…”.

[Editors' note: further revisions were requested prior to acceptance, as described below.]

Reviewer #2:

*1) I remain concerned about the threshold of 1.3-fold change. An increase in activity of 1.3-fold by a mutation “is considered significant.” The authors say that clones displaying a 1.3-fold improvement are generally verified as authentically displaying an increase in activity upon repeat examination. There are several issues about which I am concerned here. The first is that no data are provided to support this justification for the threshold; in fact, it is never articulated in the paper, although it is found in the letter. Thus the 1.3-fold threshold remains unjustified in the text. Second, the use of this threshold in defining examples of epistasis strikes me as problematic. Mutations that produce an observed increase in activity by >1.3x in one background but not another are judged to be epistatically modified. Thus, an increase of 1.4x in one background but 1.2x in another exhibit epistasis, because they are significantly improved in the first case but not the second. The authors do not present any way to establish whether the differences between these two genotypes are really statistically or biologically significant, so the claim of epistasis remains unjustified. The authors argue that a statistical analysis shows that most cases of 1.3x improvement are statistically significant (also not shown), but this is beside the point; the relevant issue is whether the* difference *between the effects of a mutation in two genetic backgrounds is statistically and biologically significant, not whether a 1.3-fold effect one mutation is different from no effect. Many, although not all, of the authors' examples of epistasis in the paper are, in fact, rather subtle examples of phenomena like this – not sign epistasis, or very large effects vs. very small or no effects, but the difference between an effect slightly greater than 1.3 and one slightly less. I therefore believe that the authors need to strengthen this aspect of the analysis. I recommend a test of differences of effect between backgrounds*.

We now include all statistics for each figure in [Supplementary-material SD6-data] (Tables B-G) and are giving *p*-values for the effect of all mutations compared to the respective background in which they occur. Where applicable, we have also calculated *p*-values for the difference in effect of a certain mutation in two genetic backgrounds to support our claim of epistasis, as suggested by the reviewer. This information can be found in four new source data files ([Supplementary-material SD1-data], [Supplementary-material SD2-data SD3-data], [Supplementary-material SD4-data]). Our analysis shows that out of 144 mutations, only six show a >1.3-fold effect, but are statistically not significant compared to the parent. When comparing the effect of a mutation in two different backgrounds, only two mutations are not statistically significant. These eight mutations are mentioned both in the relevant table and in the relevant figure legend (please see the new figure supplements as well as changes to the figure legends). We have remade all relevant figures according to the new analysis and would like to note the following changes: In Figure 4 (comparison of mutations in the *wt*- and *neo*PTE backgrounds), *t*341I has moved from panel B (neutral in *wt*PTE, deleterious in *neo*PTE) to panel A (neutral in both backgrounds) and *a*204G has moved from panel B to D (deleterious in both backgrounds).

In Figure 6, we have removed the R111*s* mutation as an example for sign epistasis, because its effect is statistically not significant in one of the two backgrounds considered. Furthermore, we now explain the 1.3-fold cut-off in the main text.

*2) The authors make inferences about clashes on a sub-angstrom scale from structures that are only at 1.6 to 2 angstrom resolution. They say that the B-factors are low, but this is of limited relevance. This indicates that there is limited thermal motion for these atoms, but the resolution of the structure still does not allow location of the atom with precision greater than the stated resolution, which is necessary to strongly support the claims made. I believe this could be dealt with in the manuscript with more direct cautionary language and softening of the claims*.

Respectfully, we disagree with this comment. We believe that the reviewer has mistaken precision with resolution. Resolution refers the ability to fully distinguish between atoms, i.e. the density from two metal ions separated by 3.3 Å will be fully separated at resolutions lower than this, e.g*.* 1.9 Å. But this does not mean that the position of each metal ion is equally likely to be anywhere within 1.9Å of its modeled position. The precision of the atomic locations is best measured by the dispersion precision indicator (described in detail in (16)). The B-factors are actually of direct relevance – in macromolecular crystallography, thermal motion, crystallographic disorder, even model building errors, all contribute to the B-factor. Moreover, the DPI directly relies on the B-factor and Rfree in its calculation. The greater the accuracy of the model (Rfree) and the lower the atomic disorder (B-factor), the more precise the atomic precision. In this case, the overall DPI for the structures of R0 (0.09), R22 (0.07), and Rev12 (0.02) are all well within the distance change that we observe in these structures. We have included the comment:

“However, the dispersion precision indicator (DPI; [16]) for each of these structures is less than a tenth of an angstrom, meaning that the observed distance changes (including the 0.5 Å shift in the metal position are significant”.

*3) The idea of different protein “species” that are “genetically incompatible” with each other is still obscure, and I do not find it persuasive. Do the authors mean that if the alleles were brought together in a heterozygote that recombination within the gene would produce nonfunctional proteins and might be selected against? Are they arguing that this might contribute to speciation? If that's what the authors mean, they should say so, but it seems very far-fetched and marginally relevant to our understanding of protein evolution. I think the effects of epistasis on the evolutionary potential of a protein are a much more solid basis for the authors' interpretation*.

To address the concerns of both Reviewers 2 and 3, we have removed the idea of “molecular speciation” in all instances and only talk about “genotypic incompatibility” between the *wt*- and *neo*PTE sequences.

It is our impression that the reviewers object only to the use of the word “speciation”, but not “incompatibility”, a term which has been applied to the level of protein sequences as discussed in our previous reply (*“*…the term “incompatibility” has been expanded to include the protein level by the community (e.g. “protein sequence incompatibility” (56), “incompatible mutations” (37), “Dobzhansky-Mueller incompatibility*”* (33))”*.* Therefore, we believe that “genotypic incompatibility”, as we call it, adequately describes the relationship between the different PTE sequences.”). We have also taken up the suggestion from Reviewer 2’s previous comments to refer to the “Dobzhansky-Muller effect” as a metaphor in one instance (Introduction, see below). We have made substantial changes throughout the text and changed the title to: “Reverse evolution leads to genotypic incompatibility despite functional and active-site convergence”.

Reviewer #3:

1) The “speciation” concept that another reviewer and I both objected to is still present in the title, Abstract and text. I think alternative wording should be used whenever possible, and the title should be changed.

Please see our reply to Reviewer 2.

*2) The Introduction is still somewhat grandiose. In particular, I am bothered by the first sentence. I do not see how reversibility relates to the question of whether re-playing life's tape would lead to the same outcome. The latter question addresses whether the same things would happen again starting from the same place, while the former asks whether an individual trajectory can be reversed. Either omit or rephrase/expand to clarify the connection*.

We have rephrased the first sentence of the Introduction.

*3) Is there any reason not to plot* k_*cat*_/K_*M*_
*values in*
Figure 1*? They would be more relevant in terms of the changes in the two enzyme activities as the evolution proceeds*.

The reason we consistently use lysate activities throughout the paper is that this is what was the basis for the selection of improved variants in our screens and therefore more adequately reflects “fitness” in our model evolutionary system, as mentioned in the text (subsection “Phenotypic reversibility in the laboratory evolution of PTE”).

In Figure 1—figure supplement 1, we show that the development of activities measured with purified enzyme at a constant enzyme and substrate concentration correlates well with the lysate data. In [Supplementary-material SD6-data], we give the *k*_*cat*_/*K*_*M*_ values for all variants shown in Figure 1, which are also in agreement with the development of lysate activities. If the reviewer feels that these values need to occupy a more prominent position in the paper, we are happy to add another figure supplement to Figure 1 equivalent to panel 1B but showing *k*_*cat*_/*K*_*M*_ values.

*4) The section head in the Results section states that “The active site converged to its original state in the reverse evolution”. I feel that this is misleading. The active site residues were not restored, and the shape of the active site is rather different. It would be correct to say that a functional active site had been restored, but I don't agree with the claim in this section (and elsewhere in the manuscript) that the active site converged to its original state. Further, the authors say that “the naphthyl binding pocket remains intact […] which likely explains why neoPTE is still bi-functional”. This statement is not consistent with the statement that the active site converged to its original state*.

We agree with the reviewer that our wording was imprecise. We have modified the text to show that the active site converges only “towards”, not “to” its original state and that this convergence is only in the key elements required for phosphotriesterase activity, not arylesterase activity.

*5) In the aforementioned subsection, it is claimed that: “the number of accessible mutational trajectories that lead to a wild-type level fitness peak from AE are highly limited”. I don't believe this statement is justified. There may well be many trajectories that could lead to* wt*-level activity that start from states that were discarded during the evolution because they were not the best variants at that particular stage*.

We thank the reviewer for pointing this out and have modified the section in question (“This failure to reach wild-type activity levels […] through less improved intermediates, may exist”).

6) A picture showing smooth and rugged fitness landscapes would be useful for readers who are not experts in molecular evolution.

Ruggedness of the fitness landscape is certainly one of our conclusions but not entirely central to our theme. Moreover, it is hard to provide a schematic view of what we observe in our experiment (reversibility and incompatibility), and we do not believe that an oversimplified scheme of a smooth vs. a rugged fitness landscape would fit our story well, but instead may be confusing and misleading. We cited a number of reviews on the topic that non-expert readers can follow (Introduction).

*7) Subsection “Emergence of incompatibility between the two seemingly identical enzymes”: the heading of this section refers to “two seemingly identical enzymes”. The enzymes are not seemingly identical, as* neo*PTE still has fairly robust arylesterase activity*.

We have changed this to say: “Emergence of incompatibility between the two PTEs”. We have also rephrased the following sentence to say: “Next, we set out to answer the question how the two enzymes exhibiting identical phosphotriesterase activity, *wt*- and *neo*PTE, are connected on the adaptive landscape.”

*8) The terminology used to describe mutations in the discussion of*
Figures 5 and 6
*is problematic. Usually the authors use the conventional notation; e.g. AxxB, which means that A at position xx was changed to B. But sometimes they use the reverse. For example, in the subsection “Mutational epistasis underlies genotypic irreversibility”, the text says: “some mutations initially increased (I172*t *and F271*l*) or were neutral (V130*l*) to phosphotriesterase activity…”. The mutations during the forward evolution were actually* t*172I,* l*271F and* l*130V. According to*
Figure 1
*and*
Figure 1—figure supplement 1*,* t*172I decreased, rather than increased, phosphotriesterase activity. This section needs to be clarified.*

We sincerely apologize that the text was still so difficult to digest and we agree that the mutations mentioned (*t*172I*, l*271F *and l*130V*)* should be written in the direction pointed out by the reviewer. To clarify our arguments, we have gone over the whole section in question (“Mutational epistasis underlies genotypic irreversibility”) and made significant changes.

*9)*
Figure 5
*is still nearly incomprehensible to me. For example, in the column labeled “R254*h*”, the legend says that the effect is calculated in the direction R254*h *in all cases. So the figure suggests that changing R254 to H increases phosphotriesterase activity by 13-fold. But the* wt *enzyme has* His *at 254, not* Arg*. So what is really meant is that changing* His*254 to* Arg *decreases phosphotriesterase activity by 13-fold, which is consistent with*
Figure 1*. Having to do these mental gymnastics for each position and in each background makes my brain hurt. I do not think this way of conveying the data is salvageable*.

We are sorry for the confusion. Figure 5 shows how the mutational effects changed in the five different backgrounds in a comprehensive manner. We believe that this is very important because all other figures only show certain parts of the complete data set. However, due to the nature of the dataset, i.e., each background has a different amino acid, the exact numbers in Figure 5 are certainly not easy to digest, although the color coding already gives a good impression of the prevalence of epistasis – this is why it is important in this figure to give all mutations in the same direction, even though this means sometimes adding a mutation, sometimes “taking it out” and calculating the effect of “putting it back in”. To better explain the numbers and directions of the effect, we rewrote the figure legend and sincerely hope will now make it easier to understand how the data was processed.

*10)*
Figure 6
*shows the effects of V130I on phosphotriesterase activity in the “forward evolution” and “AE background” contexts.*
Figure 6
*would be easier to understand if it specified V130*I *in R13*b*, and* I*172T in R5*a *(and so on), rather than using the term “forward evolution”, which initially gave me the impression that V130 was changed to I during the forward evolution. In fact, I am still confused by whether the bars at the left side of*
Figure 6
*refer to the actual mutation that occurs during the forward evolution, or the reversion of the mutation at position 130. Even after several tries, I am unable to follow the discussion in subsection “Mutational epistasis underlies genotypic irreversibility”*.

We apologize again for the confusion and have added an additional explanation to the legend of Figure 6.

We have also attempted to clarify the discussion of Figures 5 and 6 as described in our reply to comment no. 8. We have also taken up the reviewer’s suggestion to add the rounds of occurrence to the “forward evolution” variants in Figure 6 and in the text.

[Editors' note: further revisions were requested prior to acceptance, as described below.]

*An issue raised by several of the reviewers during both rounds of revision related to the terms ‘molecular speciation’ and ‘incompatibility’. While you have removed the confusing former term, the latter still appears in several places in the text. This is fine, but it needs, in some instances, a bit of clarification and adjustment of the language. For instance, the Abstract states that the reverse evolution experiment performed led “to a different sequence incompatible with the original one, despite functional identity.” As written, the statement is ambiguous and confusing in the sense that what you really mean is that certain substitutions that occurred during the reverse evolution process are incompatible with some of the residues present in the ancestral/original protein. It's not that the whole sequence is somehow incompatible with the original one, as written. Please adjust the language throughout the text where this concept arises to improve precision and meaning*.

We agree with your concern about our description of “incompatibility”. We have revised the text to clarify the meaning of the term, in particular in the Abstract, Results (subsection ““Emergence of incompatibility between the two PTEs” and Discussion).